

**Geochemistry of the dissolved loads of rivers in Southeast Coastal Region, China:**
**Anthropogenic impact on chemical weathering and carbon sequestration**
Wenjing Liu[1,2,3], Zhifang Xu[1,2,3*], Huiguo Sun[1,2,3], Tong Zhao[1,2,3], Chao Shi[1,2], Taoze Liu[4]
[1] Key Laboratory of Cenozoic Geology and Environment, Institute of Geology and
Geophysics, Chinese Academy of Sciences, Beijing 100029, China
[2] Institutions of Earth Science, Chinese Academy of Sciences, Beijing 100029, China
[3] University of Chinese Academy of Sciences, Beijing 100049, China
[4] State Key Laboratory of Environmental Geochemistry, Institute of Geochemistry, Chinese
Academy of Sciences, Guiyang, Guizhou 550002, China
* Corresponding author. zfxu@mail.iggcas.ac.cn (Zhifang Xu, Tel: +86 10 82998289)



**Abstract**:
Southeast coastal region is the most developed and populated area in China.
Meanwhile, it has been the most severe acid rain impacted region for many years. The
chemical compositions and carbon isotope ratio of dissolved inorganic carbon
($\delta^{13}C_{DIC}$) of rivers were investigated to evaluate the chemical weathering and
associated atmospheric $CO_2$ consumption rates. Mass balance calculation indicated
that the dissolved loads of major rivers in the Southeast Coastal Rivers Basin
(SECRB) were contributed by atmospheric (14.4%, 6.6-23.4%), anthropogenic
(17.8%, 0-55.2%), silicate weathering (38.3%, 10.7-74.0%) and carbonate weathering
inputs (29.4%, 3.9-62.0%). The silicate and carbonate chemical weathering rates for
these river watersheds were 10.0-29.6 t $km^{-2}$ $a^{-1}$ and 1.0-54.1 t $km^{-2}$ $a^{-1}$, respectively.
The associated mean $CO_2$ consumption rate by silicate weathering for the whole
SECRB were $167\times10^3$ mol $km^{-2}$ $a^{-1}$. The chemical and $\delta^{13}C_{DIC}$ evidences indicated
that sulfuric acid (mainly from acid deposition) was significantly involved in chemical
weathering of rocks. The calculation showed an overestimation of $CO_2$ consumption
at $0.19\times10^{12}$ g C $a^{-1}$ if sulfuric acid was ignored, which accounted for about 25% of
the total $CO_2$ consumption by silicate weathering in the SECRB. This study
quantitatively highlights that the role of sulfuric acid in chemical weathering,
suggesting that acid deposition should be considered in studies of chemical
weathering and associated $CO_2$ consumption.
**Keywords:** Southeast Coastal Rivers Basin; Chemical weathering; $CO_2$ consumption;
acid deposition;



## 1. Introduction

Chemical weathering of rocks is a key process that links geochemical cycling of solid earth to the atmosphere and ocean. It provides nutrients to terrestrial and marine ecosystems and regulates the level of atmospheric $CO_2$. As a net sink of atmospheric $CO_2$ on geologic timescales, estimation of silicate chemical weathering rates and the controlling factors are important issues related to long-term global climate change (e.g. Raymo and Ruddiman, 1992; Négrel et al. 1993; Berner and Caldeira, 1997; Gaillardet et al., 1999; Kump et al., 2000; Amiotte-Suchet et al., 2003; Oliva et al., 2003; Hartmann et al., 2009; Moon et al., 2014). As an important component in the Earth's Critical Zone (U.S. Nat. Res. Council Comm., 2001), river serves as an integrator of various natural and anthropogenic processes and products in a basin, and a carrier transporting the weathering products from continent to ocean. Therefore, the chemical compositions of river are widely used to evaluate chemical weathering and associated $CO_2$ consumption rates at catchment and/or continental scale, and examine their controlling factors (e.g., Edmond et al., 1995; Gislason et al., 1996; Galy and France-Lanord, 1999; Huh, 2003; Millot et al., 2002, 2003; Oliva et al., 2003; West et al., 2005; Moon et al., 2007; Noh et al., 2009; Shin et al., 2011; Calmels et al., 2011).

With the intensification of human activities, human perturbations to river basins have increased in frequency and magnitude. It is important to understand how such perturbations function on the current weathering systems and to predict how they will affect the Critical Zone of the future (Brantley and Lebedeva, 2011). In addition to $CO_2$, other sources of acidity (such as sulfuric, nitric and organic acids) can also

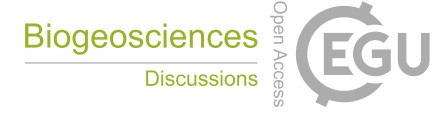



produce protons. These protons react with carbonate and silicate minerals, thus
enhance rock chemical weathering rate and flux compared with only considering
protons deriving from $CO_2$ dissolution (Calmels et al., 2007; Xu and Liu, 2010). The
effect of other sourced proton (especially $H^+$ induced by $SO_2$ and $NO_X$ coming from
anthropogenic activities) on chemical weathering is documented to be an important
mechanism modifying atmospheric $CO_2$ consumption by rock weathering (Galy and
France-Lanord, 1999; Semhi, et al., 2000; Spence and Telmer, 2005; Xu and Liu,
2007; Perrin et al., 2008; Gandois et al., 2011). Anthropogenic emissions of $SO_2$ was
projected to provide 3 to 5 times greater $H_2SO_4$ to the continental surface than the
pyrite oxidation originated $H_2SO_4$ (Lerman et al., 2007). Therefore, increasing acid
precipitation due to stronger human activities nowadays could make this mechanism
more prominently.

The role of acid precipitation on the chemical weathering and $CO_2$ consumption

has been investigated in some river catchments (Amiotte-Suchet et al., 1995; Probst et
al., 2000; Vries et al., 2003; Lerman et al., 2007; Xu and Liu, 2010). It has been
documented that silicate rocks were more easily disturbed by acid precipitation during
their weathering and soil leaching processes, because of their low buffeting capacity
(Reuss et al., 1987; Amiotte-Suchet et al., 1995). According to Amiotte-Suchet et al.
(1995), especially where crystalline rocks outcrop, the disturbance can be very high
and can induce a decrease of $CO_2$ consumption by weathering at least 73% in the
Strengbach catchment (Vosges Mountains, France). This highlights the importance of
exploring anthropogenic impact on chemical weathering and $CO_2$ consumption under



different background (e.g. lithology, climate, human activity intensity, and basin
scale) for better constraining and estimation of acid precipitation effect on rock
weathering. Asia, especially East Asia, is one of the world's major sulfur emissions
areas. However, the effect of acid precipitation on silicate weathering and associated
$CO_2$ consumption was not well evaluated in this area, especially lack of quantitative
studies.

Southeast coastal region of China is the most highly developed and

populated area in China, dominated by Mesozoic magmatic rocks (mainly granite and
volcanic rocks) in lithology. Meanwhile, it is also seriously impacted by acid rain,
with a volume-weighted mean value of pH lower than 4.5 for many years (Wang et al.,
2000; Larssen and Carmichael, 2000; Zhao, 2004; Han et al., 2006; Larssen et al.,
2006; Zhang et al., 2007a; Huang et al., 2008; Xu et al., 2011). Therefore, it is an
ideal area for evaluating silicate weathering and the effect of acid rain. In this study,
the chemical and carbon isotope composition of rivers in this area were first
systematically investigated, in order to: (i) decipher the different sources of solutes
and to quantify their contributions to the dissolved loads; (ii) calculate silicate
weathering and associated $CO_2$ consumption rates; (iii) evaluate the effects of acid
deposition on rock weathering and $CO_2$ consumption flux.
**2. Natural setting of study area**

Southeast coastal region of China, where the landscape is dominated by

mountainous and hilly terrain, has rugged coastlines and develop numerous small and
medium river system. Rivers in this region flow eastward or southward and finally





inject into the East China Sea or the South China Sea (Fig. 1), and they are
collectively named as 'Southeast Coastal Rivers' (SECRs). Large rivers are listed here
from north to south, they are: Qiantang, Cao'e, Ling, and Ou in Zhejiang province;
Jiaoxi, Huotong, Ao, Min, Jin, and Jiulong in Fujian province; Han and Rong in
Guangdong province.
The Southeast Coastal Rivers Basin (SECRB) belongs to the warm and humid
subtropical oceanic monsoon climate. The average annual temperature and
precipitation are 17-21℃ and 1400-2000 mm, respectively. The precipitation mainly
happens during May to September, and the minimum and maximum temperature
often occurs in January and July. This area is one of the most developed areas in
China, with a population more than 190 million (mean density of ~470
individuals/km$^2$), but the population mainly concentrated in the coastal urban areas.
The vegetation coverage of these river basins is more than 60%, mainly subtropical
evergreen-deciduous broadleaf forest and mostly distributing in mountains area.
Cultivated land, and industries and cities are mainly located in the plain areas and
lower reach of these rivers.
Geologically, three regional-scale fault zones are distributed across the SECRB
region (Fig. 1). They are the sub-EW-trending Shaoxing-Jiangshan fault zone, the
NE-trending Zhenghe-Dapu fault zone, and the NE-trending Changle-Nanao fault
zone (Shu et al., 2009). These fault zones dominate the direction of the mountains
ridgelines and drainages, as well as the formation of the basins and bay. The Zhenghe-
Dapu fault zone is a boundary line of Caledonian uplift belt and Hercynian-Indosinian



depression zone. Mesozoic magmatic rocks are widespread in the southeast coastal
region with a total outcrop area at about 240,000 km$^2$. Over 90% of the Mesozoic
magmatic rocks are granitoids (granites and rhyolites) and their volcanic counterpart
with minor existence of basalts (Zhou et al., 2000, 2006; Bai et al., 2014). These crust-
derived granitic rocks are mainly formed in the Yanshanian stage, and may have been
related to multiple collision events between Cathaysia and Yangtze blocks. Among the
major river basins, the proportions of magmatic rocks outcrop are about 36% in
Qiantang river basin, above 80% in Ou, Jiaoxi and Jin river basins, and around 60% in
Min, Jiulong, Han and Rong river basins (Shi, 2014). The overlying Quaternary
sediment in this area is composed of brown-yellow siltstones but is rarely developed.
The oldest basement complex is composed of metamorphic rocks of greenschist and
amphibolite facies. Sedimentary rocks categories into two types, one is mainly
composed by red clastic rocks which cover more than 40,000 km$^2$ in the study area;
the other occurs as interlayers within volcanic formations, including varicolored
mudstones and sandstones. They are mainly distributed on the west of Zhenghe-Dapu
fault zone (FJBGRM, 1985; ZJBGMR, 1989; Shu et al., 2009).
**3. Sampling and analytical method**
Water samples were collected during the high-flow season of 2010 (sample
number and locations are shown in Fig. 1). 2-L water samples were collected in the
middle channel of the river from bridges or ferries, or directly from the center of some
shallow streams in the source area. The lower reaches sampling sites were selected
distant away from the estuary to avoid the influence of seawater. Temperature (T), pH

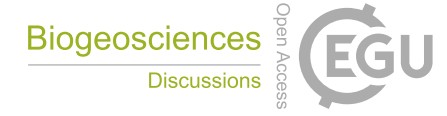

and electrical conductivity (EC) were measured in the field with a portable EC/pH
meter (YSI-6920, USA). All of the water samples for chemical analysis were filtered
in field through 0.22 μm Millipore membrane filter, and the first portion of the
filtration was discarded to wash the membrane and filter. One portion filtrate were
stored directly in HDPE bottles for anion analysis and another were acidified to pH
<2 with 6M double sub-boiling distilled $HNO_3$ for cation analysis. All containers were
previously washed with high-purity HCl and rinsed with Milli-Q 18.2 MΩ water.

Alkalinity was titrated with 0.005M HCl within 12 h after sampling. Cations

($Na^+$, $K^+$, $Ca^{2+}$ and $Mg^{2+}$) were determined using Inductively Coupled Plasma Atomic
Emission Spectrometer (ICP-AES) (IRIS Intrepid II XSP, USA). Anions ($Cl^-$, $F^-$,
$NO_3^-$ and $SO_4^{2-}$) were analyzed by ionic chromatography (IC) (Dionex Corporation,
USA). Dissolved silica was determined by spectrophotometry using the molybdate
blue method. Reagent and procedural blanks were measured in parallel to the sample
treatment, and calibration curve was evaluated by quality control standards before,
during and after the analyses of each batch of samples. Measurement reproducibility
was determined by duplicated sample and standards, which showed ±3% precision for
the cations and ±5% for the anions.

River water samples for carbon isotopic ratio ($\delta^{13}C$) of dissolved inorganic

carbon (DIC) measurements were collected in 150 ml glass bottles with air-tight caps
and preserved with $HgCl_2$ to prevent biological activity. The samples were kept
refrigerated until analysis. For the $\delta^{13}C$ measurements, the filtered samples were
injected into glass bottles with phosphoric acid. The $CO_2$ was then extracted and





cryogenically purified using a high vacuum line. $\delta^{13}$C isotopic ratios were analyzed on
Finnigen MAT-252 stable isotope mass spectrometer at the State Key Laboratory of
Environmental Geochemistry, Chinese Academy of Sciences. The results are
expressed with reference to VPDB, as follows:

$\delta^{13}\text{C} = [((^{13}\text{C}/^{12}\text{C})_{sample} / (^{13}\text{C}/^{12}\text{C})_{standard}) - 1] \times 1000$         (1)

The $\delta^{13}$C measurement has an overall precision of 0.1‰. A number of duplicate
samples were measured and the results show that the differences were less than the
range of measurement accuracy.

**4. Results**

The major parameter and ion concentrations of samples are given in Table 1. The

pH values of water samples ranged from 6.50 to 8.24, with an average of 7.23. Total
dissolved solids (TDS) of water samples varied from 35.3 to 205 mg l$^{-1}$, with an
average of 75.2 mg l$^{-1}$. Comparing with the major rivers in China, the average TDS
was significantly lower than Changjiang (224 mg l$^{-1}$, Chetelat et al., 2008), Huanghe
(557 mg l$^{-1}$, Fan et al., 2014) and Zhujiang (190 mg l$^{-1}$, Zhang et al., 2007b).
However, the average TDS was comparable to the rivers draining silicate rock
dominated areas, e.g. the upper Ganjiang (63 mg l$^{-1}$, Ji and Jiang, 2012), the Amur (70
mg l$^{-1}$, Moon et al., 2009), Xishui (101 mg l$^{-1}$, Wu et al., 2013), and north Han river in
South Korea (75.5 mg l$^{-1}$, Ryu et al., 2008). Among the major rivers in the SECRB,
the Qiantang river had the highest TDS value (averaging at 121 mg l$^{-1}$), and the Ou
river had the lowest TDS value (averaging at 48.8 mg l$^{-1}$).

Major ion compositions are shown in the cation and anion ternary diagrams (Fig.





2a and b). In comparison with rivers (e.g. the Wujiang and Xijiang) draining
carbonate rocks dominated area (Han and Liu, 2004; Xu and Liu, 2010), these rivers
in the SECRB have distinctly higher proportions of $Na^+$, $K^+$, and dissolved $SiO_2$. As
shown in the Fig. 2, most samples have high $Na^+$ and $K^+$ proportions, with an average
higher than 50% (in $\mu mol\ l^{-1}$) of the total cations, except for samples from the
Qiantang river. The concentrations of $Na^+$ and $K^+$ range from 43.5 to 555 $\mu mol\ l^{-1}$ and
42.9 to 233 $\mu mol\ l^{-1}$, with average values of 152 and 98 $\mu mol\ l^{-1}$, respectively. The
concentrations of dissolved $SiO_2$ range from 98.5 to 370 $\mu mol\ l^{-1}$, with an average of
212 $\mu mol\ l^{-1}$. $Ca^{2+}$ and $Mg^{2+}$ account for about 38% and 11.6%of the total cation
concentrations. $HCO_3^-$ is the dominant anion with concentrations ranging from 139 to
1822 $\mu mol\ l^{-1}$. On average, it comprises 60.6% (36-84.6%) of total anions on a molar
basis, followed by $SO_4^{2-}$ (14.6%), $Cl^-$ (13.1%) and $NO_3^-$ (11.8%). The major ionic
compositions indicate that water chemistry of these rivers in the SECRB is controlled
by silicate weathering. Meanwhile, it is also influenced by carbonate weathering,
especially in the Qiantang river system.

The $\delta^{13}C$ of dissolved inorganic carbon in the rivers of the SECRB are given in

Table 1. The $\delta^{13}C$ of the water samples show a wide range, from -11.0‰ to -24.3‰
(average -19.4‰), and with majority falling between -15 of -23‰. The values are
similar to most rivers draining Deccan Traps (Das et al., 2005).
**5. Discussion**

The dissolved solids in river water are commonly from atmospheric and

anthropogenic inputs and weathering of rocks within the drainage basin. It is



necessary to quantify the contribution of different sources to the dissolved loads
before deriving chemical weathering rates and associated $CO_2$ consumption.
*5.1 Atmospheric and anthropogenic inputs*
To evaluate atmospheric inputs to river waters, chloride is the most common
used reference. Generally, water samples that have the lowest $Cl^-$ concentrations are
employed to correct the proportion of atmospheric inputs in a river system (Négrel et
al., 1993; Gaillardet et al., 1997; Viers et al., 2001; Xu and Liu, 2007). In pristine
areas, the concentration of $Cl^-$ in river water is assumed to be entirely derived from
the atmosphere, provided that the contribution of evaporites is negligible. In the
SECRB, the lowest $Cl^-$ concentration was mainly found in the headwater of each
river. According to the geologic setting, no salt-bearing rocks was found in these
headwater area (FJBGRM, 1985; ZJBGMR, 1989). In addition, these areas are mainly
mountainous and sparsely populated. Therefore, we assumed that the lowest $Cl^-$
concentration of samples from the headwater of each major river came entirely from
atmosphere.
The proportion of atmosphere-derived ions in the river waters can then be
calculated by using the element/Cl ratios of the rain. Chemical compositions of rain in
the studied area have been reported at different sites, including Hangzhou, Jinhua,
Nanping, Fuzhou and Xiamen (Zhao, 2004; Zhang et al., 2007a; Huang et al., 2008;
Cheng et al., 2011; Xu et al., 2011) (Fig. 1). The volume-weighted mean
concentration of ions and Cl-normalized molar ratios are compiled in Table 2.
According to this procedure, 6.6-23.4% (averaging 14.4%) of total dissolved cations



in the major rivers of the SECRB originated from rain. Among the anions, $SO_4^{2-}$ and
$NO_3^-$ in the rivers are mainly from the atmospheric input, averaging at 74.7% for
$SO_4^{2-}$ and 68.6% for $NO_3^-$, respectively.

As the most developed and populated areas in China, the chemistry of rivers in

the SECRB could be significantly impacted by anthropogenic inputs. $Cl^-$, $NO_3^-$ and
$SO_4^{2-}$ are commonly associated with anthropogenic sources and have been used as
tracers of anthropogenic inputs in watershed. High concentrations of $Cl^-$, $NO_3^-$ and
$SO_4^{2-}$ can be found at the lower reaches of rivers in the SECRB, and an obvious
increase after flowing through plain areas and city. This tendency indicates that river
water chemistry is affected by anthropogenic inputs while passing through the
catchments. After correcting for the atmospheric contribution to river waters, the
following assumption is needed to quantitatively estimate the contributions of
anthropogenic inputs. That is, $Cl^-$ originates from only atmospheric and anthropogenic
inputs, the excess of atmospheric $Cl^-$ is regarded to present anthropogenic inputs and
balanced by Na.
*5.2 Chemical weathering inputs*

Water samples were displayed on a plot of Na-normalized molar ratios (Fig. 3).

The values of the world's large rivers (Gaillardet et al. 1999) are also shown for
comparison in the figure. A best correlations between elemental ratios were observed
for $Ca^{2+}/Na^+$ vs. $Mg^{2+}/Na^+$ ($R^2 = 0.95$, $n = 120$) and $Ca^{2+}/Na^+$ vs. $HCO_3^-/Na^+$ ($R^2 =$
0.98, $n = 120$). The samples cluster on a mixing line mainly between silicate and
carbonate end members, closer to the silicate end-member, and with little evaporite

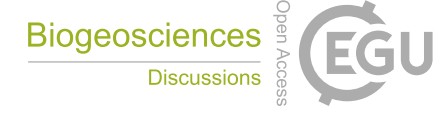



contribution. This corresponds with the distribution of rock types in the SECRB. In
addition, all water samples have equivalent ratios of $(Na^{+}+K^{+})/Cl^{-}$ larger than one,
indicating silicate weathering as the source of $Na^{+}$ and $K^{+}$ rather than chloride
evaporites dissolution.

The geochemical characteristics of the silicate and carbonate end-members can

be deduced from the correlations between elemental ratios and referred to literature
data for catchments with well-constrained lithology. After correction for atmospheric
inputs, the $Ca^{2+}/Na^{+}$, $Mg^{2+}/Na^{+}$ and $HCO_3^{-}/Na^{+}$ of the river samples ranged from 0.31
to 30, 0.16 to 6.7, and 1.1 to 64.2, respectively. According to the geological setting
(Fig. 1), there are some small rivers draining purely silicate areas in the SECRs
drainage basins. Based on the elemental ratios of these rivers, we assigned the silicate
end-member for this study as $Ca^{2+}/Na^{+}=0.41\pm0.10$, $Mg^{2+}/Na^{+}=0.20\pm0.03$ and $HCO_3^{-}$
$/Na^{+}=1.7\pm0.6$. The ratio of $(Ca^{2+}+Mg^{2+})/Na^{+}$ for silicate end-member was 0.61, which
is close to the silicate end-member of world rivers $((Ca^{2+}+Mg^{2+})/Na^{+}=0.59\pm0.17$,
Gaillardet et al., 1999). Moreover, several previous researches have documented the
chemical composition of rivers, such as the Amur and the Songhuajiang in North
China, the Xishui in the lower reaches of the Changjiang, and major rivers in South
Korea (Moon et al., 2009; Liu et al., 2013; Wu et al., 2013; Ryu et al., 2008; Shin et
al., 2011). These river basins has similar geological setting with the study area, we
could further validate the composition of silicate end-member with their results.
$Ca^{2+}/Na^{+}$ and $Mg^{2+}/Na^{+}$ ratios of silicate end-member were reported for the Amur
(0.36 and 0.22), the Songhuajiang (0.44±0.23 and 0.16), the Xishui (0.6±0.4 and



0.32±0.18), the Han (0.55 and 0.21) and six major rivers in South Korea (0.48 and
0.20) in the studies above, well bracketing our estimation for silicate end-member.

Whereas, some samples show high concentrations of $Ca^{2+}$, $Mg^{2+}$ and $HCO_3^-$,

indicating the contribution of carbonate weathering. The samples collected in the
upper reaches (Sample 12 and 13) in the Qiantang river fall close to the carbonate
end-member documented for world large rivers (Gaillardet et al., 1999). In the present
study, $Ca^{2+}/Na^+$ ratio of 0.41±0.10 and $Mg^{2+}/Na^+$ ratio of 0.20±0.03 for silicate end-
member are used to calculate the contribution of $Ca^{2+}$ and $Mg^{2+}$ from silicate
weathering. Finally, residual $Ca^{2+}$ and $Mg^{2+}$ are apportioned to carbonate weathering.
*5.3 Chemical weathering rate in the SECRBs*

Based on the above assumption, a forward model is employed to quantify the

relative contribution of the different sources to the rivers of the SECRB in this study.
(e.g. Galy and France-Lanord, 1999; Moon et al., 2007; Xu and Liu, 2007; 2010; Liu
et al., 2013). The calculated contributions of different reservoir to the total cationic
loads for large rivers and their major tributaries in the SECRB are presented in Fig. 4.
On average, the dissolved cationic loads of the rivers in the study area originate
dominantly from silicate weathering, which accounts for 38.3% (10.7-74.0%) of the
total cationic loads in molar unit. Carbonate weathering and anthropogenic inputs
account for 29.4% (3.9-62.0%) and 17.8% (0-55.2%), respectively. Contributions
from silicate weathering are high in the Ou (55.7%), Huotong (55%), Ao (48%) and
Min (48.4%) river catchments, which dominated by granitic and volcanic bedrocks. In
contrast, high contribution from carbonate weathering is observed in the Qiantang



(56.6%) and Jiulong (38.5%) river catchments. The results manifest the lithology
control on river solutes of drainage basin.

The chemical weathering rate of rocks is estimated by the mass budget, basin

area and annual discharge, expressed in ton $km^{-2}$ $a^{-1}$. The silicate weathering rate
(SWR) is calculated using major cationic concentrations from silicate weathering and
assuming that all dissolved $SiO_2$ is derived from silicate weathering (Xu and Liu,
2010), as the equation below:

$SWR = ([Na]_{sil} + [K]_{sil} + [Ca]_{sil} + [Mg]_{sil} + [SiO_2]_{riv}) \times discharge/area$        (2)

The assumption about Si could lead to overestimation of the silicate weathering

rate, as part of silica may come from dissolution of biogenic sources rather than the
weathering of silicate minerals (Millot et al., 2003; Shin et al., 2011). Thus, the
cationic silicate weathering rates ($Cat_{sil}$) were also calculated.

The carbonate weathering rate (CWR) is calculated based on the sum of $Ca^{2+}$,

$Mg^{2+}$ and $HCO_3^-$ from carbonate weathering, with half of the $HCO_3^-$ coming from
carbonate weathering being derived from the atmosphere $CO_2$, as the equation below:

$CWR = ([Ca]_{carb} + [Mg]_{carb} + 1/2[HCO_3]_{carb}) \times discharge/area$            (3)

The chemical weathering rates and fluxes are calculated for major rivers and

their main tributaries in the SECRB, and the results are shown in Table 3. Silicate and
carbonate weathering fluxes of these rivers (SWF and CWF) range from 0.02 $\times10^6$ t $a^{-1}$
to $1.29\times10^6$ t $a^{-1}$, and from $0.002\times10^6$ t $a^{-1}$ to $1.33\times10^6$ t $a^{-1}$, respectively. Among the
rivers, Min river has the highest silicate weathering flux, and Qiantang river has the
highest carbonate weathering flux. On the whole SECRB scale, $5.23\times10^6$ t $a^{-1}$ and



$4.90 \times 10^6$ t $a^{-1}$ of dissolved solids originating from silicate and carbonate weathering,
respectively, are transported into the East and South China Sea by rivers in this
region. Comparing with three largest three river basins (Changjiang, Huanghe and
Xijiang) in China, the flux of silicate weathering calculated for the SECRB is lower
than Changjiang ($9.5 \times 10^6$ t $a^{-1}$, Gaillardet et al. 1999), but higher than Huanghe
($1.52 \times 10^6$ t $a^{-1}$, Fan et al., 2014) and Xijiang ($2.62 \times 10^6$ t $a^{-1}$, Xu and Liu, 2010).
The silicate and carbonate chemical weathering rates for these river watersheds
were 10.0-29.6 t $km^{-2}$ $a^{-1}$ and 1.0-54.1 t $km^{-2}$ $a^{-1}$, respectively. The total rock
weathering rate (TWR) for the whole SECRB is 35.3 ton $km^{-2}$ $a^{-1}$, higher than the
world average (24 ton $km^{-2}$ $a^{-1}$, Gaillardet et al., 1999). The cationic silicate
weathering rates ($Cat_{sil}$) ranges from 2.4 to 10.8 ton $km^{-2}$ $a^{-1}$ for the river watersheds
in the SECRB, averaging at 6.0 ton $km^{-2}$ $a^{-1}$. Furthermore, a good linear correlation
($R^2 = 0.85$, n = 28) is observed between the $Cat_{sil}$ and runoff (Fig. 5), indicating
silicate weathering rates is controlled by the runoff as documented in previous
researches (e.g., Bluth and Kump, 1994; Gaillardet et al., 1999; Millot et al., 2002;
Oliva et al., 2003; Wu et al., 2013; Pepin et al., 2013).
*5.4 $CO_2$ consumption and the role of sulfuric acid*
To calculate atmospheric $CO_2$ consumption by silicate weathering (CSW) and by
carbonate weathering (CCW), a charge-balanced state between rock chemical
weathering-derived alkalinity and cations was assumed (Roy et al., 1999).
$$[CO_2]_{CSW} = [HCO_3]_{CSW} = [Na]_{sil} + [K]_{sil} + 2[Ca]_{sil} + 2[Mg]_{sil} \qquad (4)$$
$$[CO_2]_{CCW} = [HCO_3]_{CCW} = [Ca]_{carb} + [Mg]_{carb} \qquad (5)$$



341 The calculated $CO_2$ consumption rates by chemical weathering for each river in

342 SECRB are shown in Table 3. $CO_2$ consumption rates by carbonate and silicate

343 weathering are from 11.6 to $550\times10^3$ mol $km^{-2}$ $a^{-1}$ (averaging at $166\times10^3$ mol $km^{-2}$ $a^{-1}$)

344 and from 67.1 to $417\times10^3$ mol $km^{-2}$ $a^{-1}$ (averaging at $214\times10^3$ mol $km^{-2}$ $a^{-1}$) for major

345 river catchments in the SECRB. The regional fluxes of $CO_2$ consumption by silicate

346 and carbonate weathering is about $63.6\times10^9$ mol $a^{-1}$ ($0.76\times10^{12}$ g C $a^{-1}$) and $50.4\times10^9$

347 mol $a^{-1}$ ($0.0.60\times10^{12}$ g C $a^{-1}$) in the SECRB.

348 However, in addition to $CO_2$, $H_2SO_4$ is well documented as a significant proton

349 provider in rock weathering process (Galy and France-Lanord, 1999; Karim and

350 Veizer, 2000; Yoshimura et al., 2001; Han and Liu, 2004; Spence and Telmer, 2005;

351 Lerman and Wu, 2006; Xu and Liu 2007; 2010). Sulfuric acid can be generated by

352 natural oxidation of pyrite and anthropogenic emissions of $SO_2$ from coal combustion

353 and subsequently dissolve carbonate and silicate minerals. The consumption of $CO_2$

354 by rock weathering would be overestimated if $H_2SO_4$ induced rock weathering is

355 ignored (Spence and Telmer, 2005; Xu and Liu, 2010; Shin et al., 2011). Thus, the

356 role of sulfuric acid on the chemical weathering is crucial for an accurate estimation

357 of $CO_2$ consumption by rock weathering.

358 Rapid economic growth and increased energy command have result in severe air

359 pollution problems in China, indicated by the high levels of mineral acids

360 (predominately sulfuric) observed in precipitation (Lassen and Carmichael, 2000; Pan

361 et al., 2013; Liu et al., 2016). The national $SO_2$ emissions in 2010 reached to 30.8

362 Tg/year (Lu et al., 2011). Previous study documented that fossil fuel combustion





accounts for the dominant sulfur deposition (~77%) in China (Liu et al., 2016).
Southeast coastal region is the most severe acid rain polluted region in China, with a
volume-weighted mean value of pH lower than 4.5 for many years (Wang et al., 2000;
Larssen and Carmichael, 2000; Zhao, 2004; Larssen et al., 2006). Current sulfur and
nitrogen depositions in the Southeast coastal region are still among the highest in
China (Fang et al., 2013; Cui et al., 2014; Liu et al., 2016).

The involvement of protons originating from $H_2SO_4$ in the river waters can be

verified by the stoichiometry between cations and anions, shown in Fig. 6. In the
rivers of the SECRB, the sum cations released by silicate and carbonate weathering
were not balanced by either $HCO_3^-$ or $SO_4^{2-}$ (Fig. 6a), but were almost balanced by the
sum of $HCO_3^-$ and $SO_4^{2-}$ (Fig. 6b). This implies that both $H_2CO_3$ and $H_2SO_4$ are the
potential erosion agents in chemical weathering in the SECRB. The $\delta^{13}C$ values of the
water samples show a wide range, from -11.0‰ to -24.3‰, with an average of -
19.4‰. The $\delta^{13}C$ from soil is governed by the relative contribution from $C_3$ and $C_4$
plant (Das et al., 2005). The studied areas have subtropical temperatures and
humidity, and thus $C_3$ processes are dominant. The $\delta^{13}C$ of soil $CO_2$ is derived
primarily from $\delta^{13}C$ of organic material which typically has a value of -24 to -34‰,
with an average of -28‰ (Faure, 1986). According to previous studies, the average
value for $C_3$ trees and shrubs are from -24.4 to -30.5‰, and most of them are lower
than -28‰ in south China (Chen et al., 2005; Xiang, 2006; Dou et al., 2013). After
accounting for the isotopic effect from diffusion of $CO_2$ from soil, the resulting $\delta^{13}C$
(from the terrestrial $C_3$ plant process) should be ~ -25‰ (Cerling et al., 1991). This



mean DIC derived from silicate weathering by carbonic acid (100% from soil $CO_2$)
would yield a $\delta^{13}C$ value of -25‰. Carbonate rocks are generally derived from marine
system and, typically, have $\delta^{13}C$ value close to zero. Thus, the theoretical $\delta^{13}C$ value
of DIC derived from carbonate weathering by carbonic acid (50% from soil $CO_2$ and
50% from carbonate rocks) is -12.5‰. DIC derived from carbonate weathering by
sulfuric acid are all from carbonate rocks, thus the $\delta^{13}C$ of the DIC would be 0‰.
Based on these conclusions, sources of riverine DIC from different end-members in
the SECRB were plotted in Fig. 7. Most water samples drift away from the three
endmember mixing area (carbonate and silicate weathering by carbonic acid and
carbonate weathering by sulfuric acid) and towards the silicate weathering by $H_2SO_4$
area, clearly illustrating the effect of sulfuric acid on silicate weathering.

Considering the $H_2SO_4$ effect on chemical weathering, $CO_2$ consumption by

silicate weathering can be determined from the equation below (Moon et al., 2007;
Ryu et al., 2008; Shin et al., 2011):

$[CO_2]_{SSW} = [Na]_{sil} + [K]_{sil} + 2[Ca]_{sil} + 2[Mg]_{sil} - \gamma \times 2[SO_4]_{atmos}$          (6)

Where $\gamma$ is calculated by $cation_{sil}/(cation_{sil} + cation_{carb})$.

Based on the calculation in section 5.1, $SO_4^{2-}$ in river waters were mainly derived

from atmospheric input. Assuming sulfate in rivers derived from atmospheric input
(after correction for sea-salt contribution) are all from acid precipitation, $CO_2$
consumption rates by silicate weathering (SSW) are estimated between $31.8 \times 10^3$ mol
$km^{-2} a^{-1}$ and $363 \times 10^3$ mol $km^{-2} a^{-1}$ for major river watersheds in the SECRB. For the
whole SECRB, the actual $CO_2$ consumption rates by silicate is $167 \times 10^3$ mol $km^{-2} a^{-1}$



when the effect of $H_2SO_4$ is considered. The flux of $CO_2$ consumption is
overestimated by $15.8 \times 10^9$ mol $a^{-1}$ ($0.19 \times 10^{12}$ g C $a^{-1}$) due to the involvement of
sulfuric acid from acid precipitation, accounting for approximately 24.9% of total
$CO_2$ consumption flux by silicate weathering in the SECRB. It highlights the fact that
the drawdown of atmospheric $CO_2$ by silicate weathering can be significantly
overestimated if acid deposition is ignored in short- and long-term perspectives. The
result is important as it quantitatively shows that anthropogenic activities can
significantly affect rock weathering and associated atmospheric $CO_2$ consumption.
The quantification of this effect needs to be well evaluated in Asian and global scale
within the current and future human activity background.
**6. Conclusions**
River waters in the Southeast coastal region of China are characterized by high
proportions of $Na^+$, $K^+$ and dissolved $SiO_2$, indicating water chemistry of the rivers in
the SECRB is mainly controlled by silicate weathering. The dissolved cationic loads
of the rivers in the study area originate dominantly from silicate weathering, which
accounts for 38.3% (10.7-74.0%) of the total cationic loads. Carbonate weathering,
atmospheric and anthropogenic inputs account for 29.4% (3.9-62.0%), 14.4% (6.6-
23.4%) and 17.8% (0-55.2%), respectively. Meanwhile, more than 70% of $SO_4^{2-}$ in
the rivers derived from atmospheric input. The chemical weathering rate of silicates
and carbonates for the whole SECRB are estimated to be approximately 18.2 and 17.1
ton $km^{-2}$ $a^{-1}$. About $10.1 \times 10^6$ t $a^{-1}$ of dissolved solids originating from rock weathering
are transported into the East and South China Sea by these rivers. With the




assumption that all the protons involved in the weathering reaction are provided by
carbonic acid, the $CO_2$ consumption rates by silicate and carbonate weathering are
222 and $176 \times 10^3$ mol $km^{-2}$ $a^{-1}$, respectively. However, both water chemistry and
carbon isotope data provide evidence that sulfuric acid from precipitation serves as a
significant agent during chemical weathering. Considering the effect of sulfuric acid,
the $CO_2$ consumption rate by silicate weathering for the SECRB are $167 \times 10^3$ mol $km^{-}$
$^2$ $a^{-1}$. Therefore, the $CO_2$ consumption flux would be overestimated by $15.8 \times 10^9$ mol $a^{-}$
$^1$ $(0.19 \times 10^{12}$ g C $a^{-1})$ in the SECRB if the effect of sulfuric acid is ignored. This work
illustrates that anthropogenic disturbance by acid precipitation has profound impact
on $CO_2$ sequestration by rock weathering.
*Acknowledgements.* This work was financially supported by Natural Science
Foundation of China (Grant No. 41673020, 91747202, 41772380 and 41730857) and
the "Strategic Priority Research Program" of the Chinese Academy of Sciences (Grant
No. XDB15010405)

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



Table 1 Chemical and carbon isotopic compositions of river waters in the Southeast Coastal Rivers Basin (SECRB) of China.

| Rivers | Sample number | Date (M/D/Y) | pH | T ℃ | EC µs cm⁻¹ | Na⁺ µM | K⁺ µM | Mg²⁺ µM | Ca²⁺ µM | F µM | Cl⁻ µM | NO₃⁻ µM | SO₄²⁻ µM | HCO₃⁻ µM | SiO₂ µM | TZ+ µEq | TZ- µEq | NICB % | δ¹³C ‰ | TDS mg l⁻¹ |
|---|---|---|---|---|---|---|---|---|---|---|---|---|---|---|---|---|---|---|---|---|
| Qiantang R. | 1 | 07-8-10 | 7.42 | 28.78 | 190 | 347 | 197 | 106 | 473 | 12.0 | 303 | 62.6 | 147 | 1130 | 148 | 1703 | 1789 | -5.0 | -19.0 | 144 |
| | 2 | 07-9-10 | 7.60 | 23.84 | 146 | 87.5 | 204 | 80.9 | 496 | 11.7 | 75.2 | 124 | 121 | 907 | 156 | 1446 | 1348 | 6.7 | -19.8 | 119 |
| | 3 | 07-9-10 | 7.37 | 27.83 | 308 | 555 | 233 | 208 | 698 | 41.8 | 312 | 223 | 437 | 1170 | 170 | 2601 | 2579 | 0.9 | -17.8 | 204 |
| | 4 | 07-10-10 | 7.27 | 26.28 | 177 | 176 | 135 | 116 | 544 | 15.7 | 151 | 142 | 170 | 985 | 175 | 1632 | 1618 | 0.8 | -19.3 | 135 |
| | 5 | 07-10-10 | 7.05 | 24.15 | 123 | 130 | 101 | 66.2 | 349 | 17.7 | 94.3 | 124 | 157 | 529 | 169 | 1061 | 1061 | 0.0 | -18.7 | 91.2 |
| | 6 | 07-10-10 | 7.24 | 23.75 | 140 | 97.6 | 69.7 | 81.0 | 451 | 20.0 | 62.1 | 109 | 204 | 703 | 164 | 1231 | 1282 | -4.2 | -21.3 | 106.6 |
| | 7 | 07-11-10 | 7.40 | 23.23 | 107 | 92.5 | 70.5 | 68.3 | 327 | 14.9 | 74.9 | 104 | 147 | 486 | 156 | 954 | 960 | -0.6 | -21.0 | 82.2 |
| | 8 | 07-11-10 | 7.16 | 27.61 | 281 | 361 | 87.5 | 128 | 469 | 26.8 | 245 | 191 | 239 | 810 | 179 | 1642 | 1724 | -5.0 | -12.9 | 137.5 |
| | 9 | 07-11-10 | 7.02 | 26.48 | 140 | 275 | 120 | 60.7 | 319 | 36.2 | 199 | 150 | 180 | 437 | 236 | 1155 | 1146 | 0.8 | -13.9 | 100.2 |
| | 10 | 07-12-10 | 7.05 | 24.24 | 99 | 205 | 114 | 58.3 | 285 | 14.6 | 191 | 114 | 132 | 305 | 278 | 1005 | 874 | 13.1 | -20.9 | 85.4 |
| | 11 | 07-12-10 | 7.05 | 27.01 | 102 | 123 | 133 | 49.8 | 284 | 18.6 | 86.5 | 123 | 144 | 377 | 183 | 924 | 874 | 5.4 | -19.2 | 79.4 |
| | 12 | 07-12-10 | 7.99 | 24.18 | 260 | 50.0 | 85.4 | 212 | 993 | - | 66.8 | 153 | 235 | 1822 | 172 | 2546 | 2512 | 1.4 | -17.6 | 205.2 |
| | 13 | 07-12-10 | 7.86 | 24.59 | 231 | 43.5 | 88.4 | 189 | 859 | - | 55.1 | 97.6 | 169 | 1763 | 170 | 2228 | 2253 | -1.1 | -18.7 | 185.4 |
| | 14 | 07-12-10 | 7.69 | 22.66 | 131 | 44.1 | 81.0 | 113 | 458 | - | 19.1 | 95.2 | 107 | 920 | 143 | 1266 | 1248 | 1.4 | -18.1 | 106.8 |
| | 15 | 07-12-10 | 7.65 | 24.48 | 106 | 61.1 | 98.3 | 87.9 | 335 | - | 37.2 | 68.3 | 112 | 663 | 164 | 1005 | 992 | 1.4 | -18.6 | 87.3 |
| | 16 | 07-12-10 | 7.46 | 23.68 | 125 | 64.3 | 108 | 117 | 406 | - | 25.9 | 75.0 | 174 | 687 | 164 | 1218 | 1136 | 6.7 | -20.0 | 98.8 |
| | 17 | 07-13-10 | 7.33 | 24.08 | 139 | 59.8 | 116 | 136 | 429 | - | 29.6 | 80.4 | 209 | 752 | 162 | 1305 | 1281 | 1.9 | -20.8 | 108.1 |
| | 18 | 07-10-10 | 7.27 | 25.74 | 141 | 163 | 114 | 69.6 | 396 | 27.3 | 126 | 148 | 161 | 597 | 153 | 1209 | 1195 | 1.1 | -21.0 | 101.0 |
| Cao'e R. | 19 | 07-16-10 | 7.17 | 22.27 | 108 | 212 | 86.3 | 69.4 | 183 | 5.1 | 151 | 148 | 114 | 384 | 216 | 803 | 912 | -13.5 | -21.2 | 79.1 |
| | 20 | 07-16-10 | 7.06 | 26.57 | 182 | 401 | 77.6 | 145 | 275 | 18.3 | 269 | 185 | 245 | 534 | 215 | 1318 | 1478 | -12.2 | -20.5 | 116.9 |
| | 21 | 07-16-10 | 7.14 | 27.26 | 171 | 333 | 91.3 | 164 | 362 | 18.1 | 224 | 194 | 207 | 658 | 225 | 1475 | 1490 | -1.0 | -20.9 | 123.3 |
| | 22 | 07-16-10 | 7.08 | 27.17 | 173 | 346 | 94.4 | 168 | 364 | 18.8 | 247 | 200 | 211 | 656 | 222 | 1506 | 1526 | -1.3 | -13.0 | 125.2 |
| Ling R. | 23 | 07-15-10 | 7.07 | 24.14 | 52 | 164 | 42.9 | 34.9 | 140 | 4.9 | 40.7 | 61.5 | 68.3 | 277 | 190 | 558 | 516 | 7.6 | -12.8 | 52.1 |
| | 24 | 07-15-10 | 7.02 | 26.04 | 74 | 169 | 92.0 | 34.2 | 150 | 6.4 | 87.0 | 77.3 | 92.8 | 272 | 196 | 629 | 622 | 1.1 | -20.8 | 59.5 |
| | 25 | 07-16-10 | 7.34 | 25.03 | 92 | 159 | 80.1 | 47.3 | 235 | 19.3 | 78.0 | 71.4 | 105 | 455 | 187 | 804 | 815 | -1.4 | -22.5 | 73.9 |
| | 26 | 07-16-10 | 7.40 | 26.75 | 113 | 216 | 77.8 | 57.1 | 249 | 20.2 | 133 | 90.0 | 115 | 494 | 196 | 905 | 946 | -4.5 | -12.7 | 82.8 |
| | 27 | 07-16-10 | 7.39 | 26 | 89 | 174 | 86.4 | 56.4 | 209 | 9.0 | 99.3 | 78.4 | 99.9 | 420 | 199 | 792 | 798 | -0.8 | -14.0 | 72.7 |
| | 28 | 07-15-10 | 6.79 | 22.33 | 75 | 159 | 82.7 | 44.1 | 143 | - | 107 | 61.8 | 83.4 | 306 | 144 | 616 | 641 | -4.1 | -21.1 | 56.5 |
| | 29 | 07-15-10 | 8.24 | 27.15 | 129 | 228 | 92.1 | 83.1 | 317 | 17.2 | 177 | 90.5 | 120 | 641 | 194 | 1120 | 1148 | -2.5 | -19.2 | 97.8 |
| Ou R. | 30 | 07-13-10 | 8.08 | 28.45 | 48 | 95.2 | 107 | 38.4 | 92.1 | 15.2 | 31.8 | 43.3 | 47.4 | 291 | 221 | 463 | 461 | 0.4 | -21.7 | 50.6 |
| | 31 | 07-13-10 | 6.71 | 22.97 | 32 | 60.7 | 106 | 12.6 | 65.0 | 10.8 | 28.9 | 45.0 | 48.9 | 158 | 169 | 322 | 329 | -2.2 | -23.8 | 36.9 |
| | 32 | 07-13-10 | 7.18 | 27.59 | 73 | 107 | 127 | 36.2 | 175 | 4.3 | 57.1 | 111 | 92.0 | 283 | 210 | 655 | 634 | 3.2 | -23.4 | 62.9 |
| | 33 | 07-13-10 | 6.94 | 24.2 | 44 | 76.9 | 112 | 20.0 | 99.1 | 10.9 | 27.9 | 63.1 | 58.6 | 249 | 184 | 427 | 457 | -7.0 | -22.5 | 47.5 |
| | 34 | 07-14-10 | 7.16 | 27.45 | 90 | 187 | 127 | 41.2 | 199.5 | 17.0 | 85.6 | 102 | 116 | 367 | 251 | 796 | 787 | 1.1 | -22.4 | 76.5 |
| | 35 | 07-14-10 | 6.97 | 24.56 | 54 | 105 | 50.9 | 29.2 | 122 | 12.2 | 46.1 | 67.8 | 73.1 | 218 | 193 | 460 | 478 | -4.1 | -22.5 | 47.9 |
| | 36 | 07-14-10 | 6.82 | 21.12 | 31 | 76.4 | 133 | 12.7 | 74.5 | 7.7 | 20.7 | 36.8 | 49.1 | 192 | 162 | 383 | 348 | 9.3 | - | 39.5 |
| | 37 | 07-14-10 | 6.82 | 23.69 | 45 | 89.5 | 105 | 19.0 | 97.8 | 10.6 | 39.6 | 52.8 | 59.1 | 231 | 185 | 428 | 441 | -3.0 | -22.9 | 46.2 |
| | 38 | 07-15-10 | 6.92 | 24.69 | 37 | 100 | 89.3 | 21.1 | 49.7 | 1.7 | 36.9 | 45.5 | 52.7 | 153 | 202 | 331 | 341 | -2.9 | - | 38.9 |
| | 39 | 07-15-10 | 6.90 | 23.86 | 35 | 92.2 | 92.0 | 19.8 | 61.4 | 1.9 | 43.9 | 47.9 | 55.5 | 139 | 193 | 347 | 342 | 1.4 | -22.3 | 38.5 |
| | 40 | 07-15-10 | 7.09 | 25.56 | 47 | 117 | 112 | 25.7 | 83.4 | 8.0 | 52.4 | 63.1 | 57.4 | 232 | 193 | 447 | 462 | -3.3 | -22.5 | 48.1 |
| | 41 | 07-14-10 | 6.97 | 24.25 | 53 | 102 | 107 | 27.6 | 119 | 13.4 | 43.5 | 59.4 | 73.2 | 277 | 183 | 502 | 526 | -4.9 | -13.7 | 52.3 |





| | | | | | | | | | | | | | | | | | | | | |
|---|---|---|---|---|---|---|---|---|---|---|---|---|---|---|---|---|---|---|---|---|
| Feiyun R. | 42 | 07-17-10 | 7.28 | 25.19 | 38 | 94.0 | 81.7 | 24.0 | 75.6 | 11.4 | 59.9 | 45.7 | 51.9 | 149 | 151 | 375 | 358 | 4.5 | - | 37.2 |
| | 43 | 07-17-10 | 7.08 | 25.61 | 46 | 101 | 79.9 | 33.9 | 93.4 | 4.6 | 66.2 | 55.1 | 52.8 | 223 | 151 | 435 | 450 | -3.3 | -23.7 | 43.5 |
| Jiaoxi R. | 44 | 07-17-10 | 7.52 | 26.92 | 47 | 116 | 81.5 | 25.2 | 92.0 | 4.1 | 73.3 | 80.3 | 25.0 | 226 | 151 | 432 | 430 | 0.5 | -23.4 | 43.0 |
| | 45 | 07-17-10 | 7.45 | 27.46 | 61 | 152 | 90.2 | 34.2 | 119 | - | 136 | 59.8 | 53.5 | 238 | 184 | 548 | 542 | 1.2 | -23.1 | 51.8 |
| | 46 | 07-18-10 | 6.90 | 27.66 | 53 | 127 | 88.1 | 33.4 | 94.4 | 7.0 | 123 | 93.1 | 30.4 | 209 | 177 | 471 | 486 | -3.3 | -14.4 | 47.4 |
| Huotong R. | 47 | 07-18-10 | 7.34 | 24 | 43 | 116 | 78.8 | 26.1 | 58.4 | 5.4 | 68.7 | 49.7 | 20.1 | 197 | 190 | 364 | 355 | 2.3 | -22.8 | 39.6 |
| Ao R. | 48 | 07-19-10 | 7.24 | 31.44 | 124 | 294 | 121 | 102 | 209 | 24.3 | 204 | 73.6 | 52.0 | 717 | 370 | 1036 | 1100 | -6.1 | -19.4 | 105.4 |
| | 49 | 07-19-10 | 7.13 | 27.82 | 46 | 109 | 96.3 | 30.0 | 73.8 | - | 72.0 | 51.3 | 22.5 | 234 | 236 | 413 | 402 | 2.6 | - | 46.2 |
| | 50 | 07-18-10 | 6.98 | 28.65 | 53 | 140 | 88.4 | 40.8 | 100 | 3.0 | 82.9 | 58.6 | 20.9 | 294 | 233 | 511 | 477 | 6.6 | -22.3 | 52.2 |
| Min R. | 51 | 07-27-10 | 7.11 | 28.4 | 42 | 116 | 92.0 | 40.5 | 119 | 18.0 | 43.9 | 35.5 | 26.0 | 382 | 182 | 526 | 513 | 2.4 | -19.4 | 52.7 |
| | 52 | 07-27-10 | 7.17 | 30 | 51 | 102 | 97.9 | 41.7 | 107 | 4.6 | 29.4 | 45.3 | 35.0 | 350 | 221 | 496 | 495 | 0.2 | - | 53.3 |
| | 53 | 07-27-10 | 7.08 | 29.4 | 99 | 214 | 92.7 | 46.4 | 126 | 18.4 | 50.1 | 39.8 | 118 | 327 | 154 | 651 | 654 | -0.4 | -20.8 | 74.0 |
| | 54 | 07-27-10 | 7.06 | 29.1 | 44 | 107 | 99.6 | 28.1 | 114 | 16.4 | 18.7 | 36.4 | 44.3 | 305 | 265 | 491 | 449 | 8.5 | -17.6 | 53.6 |
| | 55 | 07-27-10 | 7.42 | 29.4 | 57 | 139 | 93.7 | 49.8 | 113 | 3.1 | 67.1 | 56.3 | 26.6 | 384 | 236 | 558 | 561 | -0.5 | -16.4 | 58.6 |
| | 56 | 07-27-10 | 7.12 | 27.8 | 51 | 103 | 91.0 | 50.8 | 106 | 4.7 | 82.8 | 35.1 | 63.5 | 249 | 225 | 507 | 494 | 2.5 | - | 51.3 |
| | 57 | 07-27-10 | 7.08 | 27.5 | 40 | 125 | 45.0 | 36.8 | 107 | 12.1 | 43.6 | 44.5 | 29.3 | 288 | 211 | 457 | 435 | 5.0 | -21.1 | 47.4 |
| | 58 | 07-27-10 | 6.99 | 27.2 | 52 | 121 | 98.0 | 42.4 | 115 | 16.7 | 87.1 | 36.6 | 70.9 | 277 | 228 | 535 | 542 | -1.4 | -11.4 | 55.3 |
| | 59 | 07-27-10 | 6.87 | 29 | 59 | 154 | 91.4 | 59.4 | 124 | 16.5 | 77.8 | 36.7 | 88.3 | 272 | 222 | 612 | 563 | 8.0 | -20.3 | 57.2 |
| | 60 | 07-27-10 | 7.31 | 27.1 | 78 | 109 | 92.1 | 59.1 | 181 | 21.2 | 123 | 37.5 | 78.4 | 355 | 202 | 682 | 672 | 1.4 | -18.7 | 63.1 |
| | 61 | 07-27-10 | 7.22 | 27.8 | 37 | 122 | 83.3 | 52.8 | 142 | 17.4 | 111 | 37.3 | 80.4 | 288 | 221 | 596 | 597 | -0.2 | -22.3 | 58.1 |
| | 62 | 07-27-10 | 7.16 | 28.1 | 58 | 104 | 83.3 | 59.3 | 163 | 24.0 | 34.6 | 34.5 | 118 | 294 | 214 | 632 | 599 | 5.2 | -13.4 | 59.5 |
| | 63 | 07-27-10 | 7.26 | 28.3 | 87 | 139 | 86.1 | 60.9 | 191 | 14.8 | 48.0 | 93.0 | 109 | 347 | 226 | 729 | 707 | 3.0 | -21.4 | 68.6 |
| | 64 | 07-27-10 | 7.00 | 28.8 | 87 | 127 | 93.1 | 58.7 | 195 | 6.6 | 59.8 | 81.1 | 60.9 | 480 | 232 | 729 | 743 | -2.0 | -11.0 | 74.0 |
| | 65 | 07-28-10 | 6.97 | 27.9 | 37 | 163 | 82.1 | 52.2 | 140 | 20.2 | 53.1 | 60.0 | 106 | 306 | 221 | 630 | 632 | -0.2 | - | 61.9 |
| | 66 | 07-13-10 | 7.07 | 27.96 | 59 | 91.9 | 110 | 40.0 | 127 | 24.8 | 62.0 | 79.3 | 62.3 | 249 | 228 | 535 | 515 | 3.8 | - | 54.8 |
| | 67 | 07-28-10 | 7.12 | 29.7 | 38 | 108 | 93.4 | 45.9 | 133 | 12.4 | 48.3 | 34.0 | 56.6 | 368 | 220 | 560 | 564 | -0.7 | - | 57.7 |
| | 68 | 07-27-10 | 7.03 | 29.9 | 62 | 128 | 96.7 | 57.6 | 148 | 23.3 | 81.6 | 36.8 | 74.1 | 374 | 203 | 635 | 641 | -0.9 | -12.4 | 61.7 |
| | 69 | 07-27-10 | 7.01 | 28.8 | 60 | 102 | 89.1 | 73.6 | 138 | 9.6 | 50.6 | 74.1 | 32.7 | 417 | 233 | 615 | 607 | 1.3 | -21.0 | 62.3 |
| | 70 | 07-27-10 | 7.06 | 26.5 | 37 | 93.5 | 93.1 | 34.7 | 87.3 | - | 26.6 | 34.8 | 37.1 | 312 | 222 | 431 | 448 | -3.9 | -13.1 | 49.1 |
| | 71 | 07-27-10 | 7.09 | 26.5 | 25 | 62.6 | 92.7 | 27.0 | 61.5 | 4.7 | 21.5 | 18.6 | 43.4 | 191 | 154 | 332 | 318 | 4.2 | -16.0 | 35.3 |
| | 72 | 07-28-10 | 7.07 | 30.1 | 39 | 76.3 | 87.9 | 35.1 | 87.6 | 7.4 | 43.1 | 36.6 | 35.5 | 266 | 175 | 409 | 416 | -1.7 | -19.4 | 43.5 |
| | 73 | 07-27-10 | 7.01 | 28.7 | 47 | 84.9 | 95.4 | 56.7 | 106 | 12.7 | 51.8 | 49.2 | 57.2 | 315 | 211 | 506 | 531 | -4.8 | - | 53.8 |
| | 74 | 07-27-10 | 6.85 | 28.7 | 50 | 93.6 | 85.9 | 52.4 | 107 | 14.1 | 62.8 | 57.5 | 57.0 | 252 | 217 | 498 | 487 | 2.2 | -19.9 | 50.9 |
| | 75 | 07-27-10 | 7.11 | 29.7 | 69 | 117 | 85.2 | 73.4 | 159 | 7.6 | 63.7 | 75.2 | 47.4 | 418 | 230 | 666 | 652 | 2.2 | -22.2 | 65.0 |
| | 76 | 07-28-10 | 6.93 | 28.9 | 59 | 112 | 88.0 | 61.8 | 122 | 6.0 | 57.4 | 89.3 | 42.0 | 349 | 224 | 568 | 580 | -2.2 | -22.0 | 58.8 |
| | 77 | 07-21-10 | 7.76 | 32.4 | 51.2 | 163 | 85.5 | 52.8 | 151 | 20.2 | 55.3 | 70.3 | 78.6 | 372 | 175 | 656 | 655 | 0.3 | -12.5 | 61.8 |
| | 78 | 07-28-10 | 7.29 | 26.8 | 106 | 129 | 75.3 | 84.0 | 321 | 24.0 | 56.2 | 41.0 | 166 | 599 | 202 | 1013 | 1028 | -1.4 | -16.3 | 90.3 |
| | 79 | 07-21-10 | 7.09 | 26.96 | 56 | 112 | 87.6 | 37.1 | 129 | 4.5 | 51.5 | 44.9 | 61.9 | 327 | 276 | 531 | 547 | -2.9 | -22.2 | 59.1 |
| | 80 | 07-21-10 | 7.64 | 33.37 | 83 | 114 | 96.2 | 60.6 | 151 | 16.7 | 53.0 | 40.6 | 102 | 371 | 242 | 633 | 670 | -5.8 | -12.8 | 66.2 |
| | 81 | 07-21-10 | 7.83 | 31.27 | 65 | 131 | 102 | 52.7 | 141 | 16.1 | 45.3 | 49.7 | 91.8 | 324 | 239 | 620 | 603 | 2.8 | -13.4 | 61.8 |
| | 82 | 07-21-10 | 6.84 | 28.35 | 66 | 132 | 101 | 52.5 | 141 | 5.8 | 63.8 | 54.1 | 91.6 | 304 | 243 | 621 | 606 | 2.5 | -22.7 | 61.5 |
| | 83 | 07-21-10 | 7.42 | 30.7 | 98 | 217 | 113 | 59.2 | 210 | 18.4 | 98.7 | 63.5 | 84.7 | 496 | 320 | 868 | 827 | 4.6 | -18.9 | 84.5 |
| | 84 | 07-27-10 | 7.26 | 26.3 | 46 | 104 | 102 | 29.7 | 121 | 3.6 | 55.2 | 51.9 | 55.5 | 294 | 193 | 507 | 512 | -0.9 | -21.6 | 51.9 |
| | 85 | 07-27-10 | 7.07 | 25.4 | 30 | 73.3 | 99.2 | 19.6 | 78.8 | - | 22.9 | 40.0 | 49.2 | 203 | 170 | 369 | 365 | 1.3 | -21.1 | 39.8 |
| | 86 | 07-27-10 | 7.50 | 27.3 | 45 | 102 | 102 | 26.5 | 114 | 2.4 | 35.1 | 39.7 | 57.2 | 260 | 217 | 484 | 449 | 7.3 | -15.7 | 49.6 |
| | 87 | 07-27-10 | 7.47 | 26.9 | 51 | 141 | 100 | 43.6 | 109 | 7.9 | 79.7 | 42.4 | 57.7 | 311 | 217 | 547 | 548 | -0.3 | -20.1 | 55.6 |





| Location | | | | | | | | | | | | | | | | | | | | |
|---|---|---|---|---|---|---|---|---|---|---|---|---|---|---|---|---|---|---|---|---|
| | 88 | 07-19-10 | 7.99 | 31.74 | 63 | 167 | 96.5 | 33.5 | 115 | 8.0 | 105 | 35.5 | 38.1 | 331 | 218 | 561 | 548 | 2.3 | -13.5 | 55.9 |
| | 89 | 07-21-10 | 6.77 | 28.19 | 65 | 132 | 93.6 | 56.0 | 145 | 15.6 | 60.6 | 78.8 | 75.4 | 333 | 243 | 627 | 624 | 0.5 | -22.6 | 63.3 |
| Jin R. | 90 | 07-27-10 | 7.36 | 25.8 | 128 | 126 | 94.8 | 88.9 | 406 | 22.9 | 51.4 | 39.4 | 229 | 595 | 208 | 1211 | 1143 | 5.6 | -20.7 | 100 |
| | 91 | 07-27-10 | 7.40 | 26.9 | 123 | 143 | 103 | 82.7 | 347 | 21.0 | 83.5 | 203 | 182 | 463 | 226 | 1105 | 1115 | -0.9 | -21.3 | 98.4 |
| | 92 | 07-27-10 | 7.00 | 27.4 | 88 | 170 | 98.8 | 56.8 | 205 | 7.2 | 137 | 117 | 106 | 327 | 205 | 793 | 792 | 0.1 | -22.5 | 71.8 |
| | 93 | 07-27-10 | 7.32 | 28.7 | 73 | 201 | 116 | 87.1 | 318 | 20.0 | 93.5 | 41.5 | 189 | 508 | 267 | 1128 | 1020 | 9.6 | -21.7 | 95.3 |
| Jiulong R. | 94 | 07-30-10 | 6.50 | 23.47 | 29 | 72.3 | 92.4 | 22.8 | 59.8 | 12.4 | 25.1 | 27.0 | 50.0 | 189 | 213 | 330 | 341 | -3.4 | -18.1 | 40.1 |
| | 95 | 07-30-10 | 7.06 | 29.35 | 120 | 136 | 96.9 | 106 | 339 | 5.1 | 67.7 | 66.3 | 249 | 469 | 202 | 1124 | 1100 | 2.1 | -20.8 | 94.2 |
| | 96 | 07-30-10 | 7.45 | 27.6 | 104 | 79.5 | 97.5 | 106 | 363 | 14.4 | 70.7 | 50.0 | 99.9 | 729 | 184 | 1116 | 1049 | 6.0 | -18.9 | 93.7 |
| | 97 | 07-31-10 | 7.36 | 26.59 | 139 | 140 | 100 | 142 | 432 | 15.5 | 79.6 | 78.3 | 274 | 573 | 196 | 1388 | 1278 | 8.0 | -19.7 | 108.8 |
| | 98 | 07-31-10 | 7.72 | 26.18 | 88 | 77.6 | 96.2 | 69.0 | 313 | 19.9 | 39.7 | 34.6 | 63.8 | 731 | 251 | 938 | 933 | 0.5 | -18.4 | 89.4 |
| | 99 | 07-30-10 | 7.43 | 26.96 | 119 | 200 | 93.8 | 100.2 | 298 | 19.9 | 122 | 80.5 | 225 | 387 | 202 | 1091 | 1040 | 4.7 | -20.5 | 89.5 |
| | 100 | 07-28-10 | 7.41 | 26.66 | 112 | 173 | 97.9 | 94.4 | 286 | 46.1 | 118 | 152 | 201 | 364 | 207 | 1033 | 1036 | -0.3 | -20.9 | 92.2 |
| | 101 | 07-29-10 | 7.16 | 29.35 | 82 | 151 | 110 | 55.4 | 178 | 4.9 | 71.2 | 170 | 53.2 | 385 | 305 | 727 | 732 | -0.7 | -21.2 | 76.1 |
| | 102 | 07-29-10 | 7.10 | 28.9 | 100 | 222 | 98.3 | 49.4 | 249 | 3.6 | 126 | 157 | 52.7 | 532 | 303 | 917 | 920 | -0.3 | -21.7 | 90.0 |
| | 103 | 07-28-10 | 7.20 | 31.15 | 138 | 339 | 111 | 81.2 | 277 | 9.2 | 280 | 285 | 88.6 | 515 | 317 | 1165 | 1256 | -7.8 | -19.0 | 112 |
| | 104 | 07-28-10 | 7.16 | 27.09 | 101 | 261 | 95.8 | 81.7 | 235 | 40.3 | 173 | 80.1 | 174 | 291 | 136 | 990 | 892 | 9.9 | -24.3 | 75.4 |
| Zhang R. | 105 | 07-28-10 | 8.08 | 30.6 | 93 | 195 | 96.1 | 61.1 | 167 | 16.8 | 157 | 193 | 55.2 | 281 | 288 | 748 | 741 | 0.9 | -21.5 | 73.8 |
| Dongxi R. | 106 | 07-28-10 | 7.20 | 30.9 | 78 | 263 | 99.0 | 41.5 | 115 | 14.5 | 238 | 65.3 | 30.0 | 283 | 309 | 675 | 646 | 4.4 | -20.8 | 66.7 |
| Huangang R. | 107 | 07-28-10 | 7.40 | 30.5 | 99 | 253 | 85.6 | 53.0 | 154 | 7.7 | 190 | 63.5 | 56.4 | 460 | 278 | 754 | 827 | -9.6 | -20.0 | 77.4 |
| Han R. | 108 | 07-31-10 | 7.31 | 27.1 | 68 | 136 | 61.5 | 45.2 | 195 | 16.1 | 37.7 | 45.3 | 93.7 | 345 | 218 | 678 | 615 | 9.2 | -21.9 | 62.0 |
| | 109 | 07-30-10 | 7.38 | 26.94 | 88 | 116 | 103 | 63.6 | 265 | 6.4 | 53.4 | 72.2 | 84.9 | 584 | 244 | 876 | 879 | -0.4 | -20.4 | 83.7 |
| | 110 | 07-30-10 | 6.66 | 25.55 | 71 | 114 | 96.2 | 47.6 | 168 | 8.0 | 56.9 | 54.6 | 143 | 230 | 203 | 642 | 628 | 2.2 | -17.9 | 59.7 |
| | 111 | 07-30-10 | 6.66 | 27.76 | 83 | 135 | 104 | 63.8 | 203 | 8.6 | 54.5 | 74.9 | 173 | 302 | 336 | 774 | 777 | -0.4 | -20.6 | 78.7 |
| | 112 | 07-30-10 | 7.31 | 30.81 | 56 | 168 | 74.0 | 39.1 | 118 | 13.5 | 62.9 | 44.4 | 81.4 | 237 | 245 | 556 | 507 | 8.8 | -21.4 | 54.6 |
| | 113 | 07-31-10 | 7.28 | 28.73 | 98 | 137 | 99.3 | 85.6 | 270 | 9.2 | 88.8 | 59.1 | 118 | 565 | 233 | 948 | 949 | -0.1 | -19.7 | 86.6 |
| | 114 | 07-31-10 | 7.27 | 31.42 | 123 | 193 | 105 | 98.2 | 319 | 20.7 | 120 | 102 | 157 | 570 | 229 | 1132 | 1107 | 2.2 | -19.7 | 98.2 |
| | 115 | 07-30-10 | 7.43 | 29.89 | 85 | 115 | 97.5 | 65.5 | 244 | 6.5 | 46.5 | 58.6 | 103 | 511 | 251 | 832 | 822 | 1.1 | -20.8 | 79.3 |
| | 116 | 07-31-10 | 7.61 | 30.98 | 99 | 123 | 104 | 85.9 | 264 | 5.6 | 58.8 | 90.9 | 108 | 588 | 98 | 926 | 952 | -2.9 | -20.0 | 79.4 |
| | 117 | 07-31-10 | 7.31 | 29.96 | 93 | 151 | 103 | 78.1 | 250 | 15.4 | 68.0 | 99.1 | 173 | 379 | 233 | 909 | 891 | 1.9 | -21.9 | 81.8 |
| | 118 | 07-31-10 | 7.35 | 28.4 | 2 | 233 | 84.2 | 101 | 323 | 12.8 | 84.0 | 101 | 203 | 460 | 229 | 1165 | 1051 | 9.8 | -21.1 | 94.7 |
| | 119 | 07-31-10 | 7.67 | 30.38 | 93 | 136 | 87.8 | 73.6 | 231 | 16.4 | 64.6 | 94.4 | 184 | 382 | 226 | 834 | 909 | -9.1 | -20.8 | 80.5 |
| Rong R. | 120 | 07-30-10 | 7.57 | 31.83 | 68 | 193 | 79.1 | 50.3 | 146 | 16.4 | 192 | 84.0 | 31.5 | 344 | 309 | 664 | 683 | -2.8 | -20.3 | 65.8 |
| | 121 | 07-30-10 | 6.96 | 30.62 | 94 | 509 | 103 | 56.1 | 213 | 15.9 | 511 | 78.5 | 82.3 | 379 | 222 | 1150 | 1133 | 1.5 | -20.0 | 94.4 |



Table 2 Chemical compositions of precipitation at different sites located within the studied area (in µmol/l and molar ratio).

| Province | Location | pH | F$^-$ | Cl$^-$ | NO$_3^-$ | SO$_4^{2-}$ | NH$_4^+$ | K$^+$ | Na$^+$ | Ca$^{2+}$ | Mg$^{2+}$ | NO$_3$/Cl | SO$_4$/Cl | K/Cl | Na/Cl | Ca/Cl | Mg/Cl | Reference |
|---|---|---|---|---|---|---|---|---|---|---|---|---|---|---|---|---|---|---|
| Zhejiang | Hangzhou | 4.5 | 5.76 | 13.9 | 38.4 | 55 | 79.9 | 4.18 | 12.2 | 26 | 3.53 | 2.76 | 3.96 | 0.3 | 0.88 | 1.87 | 0.25 | Xu et al., 2011 |
| | Jinhua | 4.54 | 9.05 | 8.51 | 31.2 | 47.6 | 81.1 | 4.73 | 6.27 | 24 | 1.73 | 3.67 | 5.59 | 0.56 | 0.74 | 2.81 | 0.2 | Zhang et al., 2007 |
| Fujian | Nanping | 4.81 | 0.8 | 5.8 | 26.6 | 18.3 | 38 | 4.9 | 5.4 | 12.9 | 2.7 | 4.59 | 3.16 | 0.84 | 0.93 | 2.22 | 0.47 | Cheng et al., 2011 |
| | Fuzhou | | 5.26 | 21.4 | 24.9 | 48.5 | 78.1 | 4.1 | 2.61 | 32.7 | 1.25 | 1.16 | 2.26 | 0.19 | 0.12 | 1.53 | 0.06 | Zhao, 2004 |
| | Xiamen | 4.57 | 15.3 | 23.7 | 22.1 | 31.3 | 37.7 | 3.58 | 36.1 | 21.5 | 4.94 | 0.93 | 1.32 | 0.15 | 1.52 | 0.91 | 0.21 | Zhao, 2004 |
| Average | | | | | | | | | | | | 2.62 | 3.26 | 0.41 | 0.84 | 1.87 | 0.24 | |

Table 3 Contribution of each reservoir, fluxes, chemical weathering and associated CO$_2$ consumption rates for the major rivers and their main tributaries in the SECRB.

| Major river | Tributaries | Location | Discharge | Area | Runoff | Contribution (%) | | | | Fluxes ($10^6$ ton a$^{-1}$) | | Weathering rate (ton km$^{-2}$a$^{-1}$) | | | | CO$_2$ consumption rate ($10^3$ mol km$^{-2}$ a$^{-1}$) | | |
|---|---|---|---|---|---|---|---|---|---|---|---|---|---|---|---|---|---|---|
| | | | $10^9$ m$^3$a$^{-1}$ | $10^3$ km$^2$ | mm a$^{-1}$ | Rain | Anth. | Sil. | Carb. | SWF | CWF | Cat$_{sil}$[a] | SWR[b] | CWR[b] | TWR[b] | CSW[c] | CCW[c] | SSW[d] |
| Qiantang | | Fuyang | 33.6 | 38.32 | 877 | 9 | 14 | 23 | 54 | 0.51 | 1.33 | 5.2 | 13.3 | 34.8 | 48.0 | 171 | 352 | 150 |
| | Fenshui | Tonglu | 3.13 | 3.43 | 913 | 7 | 14 | 18 | 62 | 0.05 | 0.19 | 5.7 | 15.3 | 54.1 | 69.3 | 173 | 550 | 158 |
| Cao'e | | Shangyu | 4.53 | 5.092 | 890 | 7 | 23 | 26 | 44 | 0.10 | 0.19 | 7.0 | 18.9 | 36.7 | 55.5 | 279 | 383 | 249 |
| Ling | | Linhai | 5.4 | 6.613 | 817 | 9 | 22 | 24 | 45 | 0.09 | 0.17 | 4.7 | 14.2 | 26.1 | 40.3 | 167 | 267 | 143 |
| Ou | | Wenzhou | 18.11 | 18.1 | 1001 | 20 | 6 | 56 | 18 | 0.32 | 0.12 | 6.6 | 17.6 | 6.5 | 24.0 | 235 | 66 | 148 |
| | Songyang | Longli | 2.03 | 1.995 | 1018 | 16 | 8 | 46 | 30 | 0.04 | 0.03 | 7.1 | 19.9 | 13.6 | 33.5 | 240 | 137 | 176 |
| | Xiaoxi | Qingtian | 3.98 | 3.405 | 1169 | 23 | 0 | 74 | 4 | 0.07 | 0.00 | 8.7 | 20.1 | 1.4 | 21.5 | 298 | 14 | 154 |
| | Nanxi | Huangtian | 2.85 | 2.49 | 1145 | 21 | 9 | 63 | 7 | 0.05 | 0.01 | 8.1 | 21.3 | 2.6 | 24.0 | 292 | 27 | 162 |
| Jiaoxi | | Saiqi | 4.0 | 5.638 | 709 | 20 | 30 | 26 | 23 | 0.06 | 0.03 | 2.4 | 10.0 | 5.4 | 15.4 | 67 | 57 | 32 |
| Huotong | | Badu | 1.804 | 2.244 | 804 | 22 | 18 | 54 | 5 | 0.03 | 0.00 | 4.0 | 13.2 | 1.0 | 14.2 | 147 | 12 | 62 |
| Ao | | Lianjiang | 2.77 | 3.17 | 874 | 17 | 17 | 48 | 17 | 0.05 | 0.02 | 5.1 | 17.3 | 5.4 | 22.7 | 188 | 56 | 122 |
| Min | | Minhou | 60.55 | 60.99 | 993 | 15 | 10 | 48 | 27 | 1.29 | 0.67 | 6.6 | 21.1 | 11.0 | 32.1 | 250 | 115 | 187 |
| | Futun | Nanping | 14.2 | 13.733 | 1034 | 15 | 14 | 49 | 22 | 0.29 | 0.13 | 7.0 | 20.8 | 9.4 | 30.2 | 268 | 100 | 195 |
| | Shaxi | Qingzhou | 11.16 | 11.793 | 946 | 13 | 9 | 42 | 36 | 0.23 | 0.19 | 6.1 | 19.3 | 15.8 | 35.1 | 230 | 162 | 182 |
| | Jianxi | Daheng | 16.4 | 16.396 | 1000 | 16 | 10 | 45 | 29 | 0.31 | 0.17 | 5.7 | 19.1 | 10.4 | 29.5 | 208 | 110 | 148 |
| | Youxi | Youxikou | 4.621 | 5.436 | 850 | 15 | 8 | 46 | 31 | 0.10 | 0.06 | 5.4 | 17.7 | 10.8 | 28.5 | 197 | 113 | 148 |
| | Dazhangxi | Minhou | 4.758 | 4.843 | 982 | 15 | 21 | 47 | 17 | 0.09 | 0.03 | 6.2 | 19.1 | 6.7 | 25.8 | 228 | 69 | 153 |
| Jin | | Nan'an | 3.65 | 3.101 | 1177 | 9 | 10 | 40 | 41 | 0.09 | 0.11 | 10.8 | 29.6 | 34.4 | 64.0 | 417 | 351 | 363 |
| | Dongxi | Honglai | 1.4 | 1.917 | 730 | 12 | 22 | 28 | 38 | 0.02 | 0.03 | 3.8 | 12.7 | 14.3 | 27.0 | 126 | 146 | 99 |
| Jiulong | | Longhai | 14.07 | 14.741 | 954 | 12 | 22 | 35 | 31 | 0.22 | 0.28 | 7.1 | 14.8 | 19.1 | 34.0 | 277 | 198 | 212 |
| | Beixi | Shajian | 9.28 | 9.64 | 963 | 13 | 14 | 28 | 45 | 0.17 | 0.26 | 5.8 | 17.8 | 27.2 | 45.0 | 211 | 281 | 168 |
| | Xi'xi | Zhangzhou | 3.657 | 3.74 | 978 | 10 | 32 | 25 | 33 | 0.09 | 0.09 | 6.6 | 25.2 | 25.3 | 50.5 | 236 | 260 | 186 |
| Zhang | | Yunxiao | 1.01122 | 1.038 | 974 | 16 | 25 | 29 | 29 | 0.02 | 0.01 | 5.1 | 21.9 | 14.1 | 36.0 | 174 | 146 | 114 |
| Dongxi | | Zhao'an | 0.958 | 1.127 | 850 | 16 | 41 | 26 | 17 | 0.02 | 0.01 | 4.0 | 19.8 | 7.0 | 26.8 | 129 | 74 | 64 |
| Huanggang | | Raoping | 1.637 | 1.621 | 1010 | 15 | 30 | 34 | 21 | 0.04 | 0.02 | 6.0 | 22.8 | 11.1 | 33.9 | 227 | 115 | 145 |
| Han | | Chaozhou | 25.4 | 30.112 | 844 | 16 | 7 | 38 | 39 | 0.51 | 0.51 | 5.4 | 16.8 | 16.9 | 33.7 | 206 | 174 | 155 |




| | | | | | | | | | | | | | | | | | |
|---|---|---|---|---|---|---|---|---|---|---|---|---|---|---|---|---|---|
| Dingjiang | Chayang | 11.5 | | 11.802 | 974 | 17 | 6 | 46 | 32 | 0.32 | 0.18 | 7.2 | 26.9 | 15.4 | 42.3 | 275 | 158 | 201 |
| Rong | Jieyang | 3.11 | | 4.408 | 706 | 10 | 55 | 11 | 24 | 0.05 | 0.07 | 2.6 | 12.0 | 14.8 | 26.8 | 67 | 152 | 46 |
| Whole SECRB | | | 270 | 287 | 939 | | | | | 5.23 | 4.90 | 6.0 | 18.2 | 17.1 | 35.3 | 222 | 176 | 167 |

[a] $Cat_{sil}$ are calculated based on the sum of cations from silicate weathering.

[b] SWR, CWR and TWR represent silicate weathering rates (assuming all dissolved silica is derived from silicate weathering), carbonate weathering rates and total weathering rates, respectively.

[c] $CO_2$ consumption rate with assumption that all the protons involved in the weathering reaction are provided by carbonic acid.

[d] Estimated $CO_2$ consumption rate by silicate weathering when $H_2SO_4$ originating from acid precipitation is taken into account.

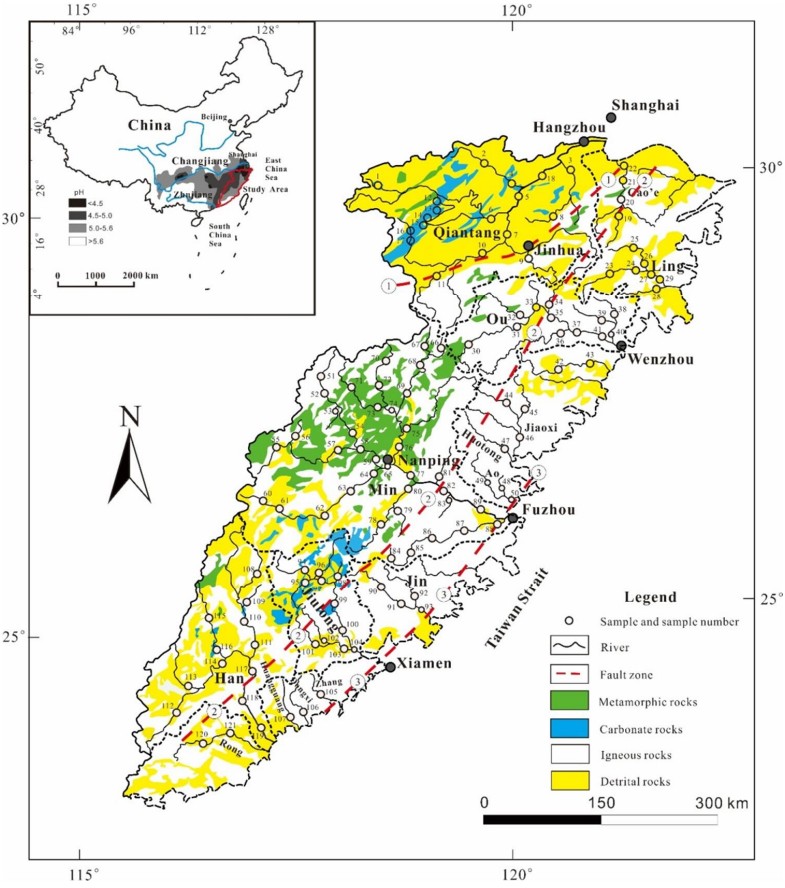

Fig. 1. Sketch map showing the lithology, sampling locations, and sample number of

the SECRs drainage basin, and regional rain water pH ranges are shown in the sketch

map at the upper-left. (modified from Zhou and Li, 2000; Shu et al., 2009; Xu et al.,

2016, rain water acidity distribution of China mainland is from State Environmental

Protection Administration of China). ①Shaoxing-Jiangshan fault zone; ②Zhenghe-

Dapu fault zone; ③Changle-Nanao fault zone. The figure was created by CorelDraw

software version 17.1.



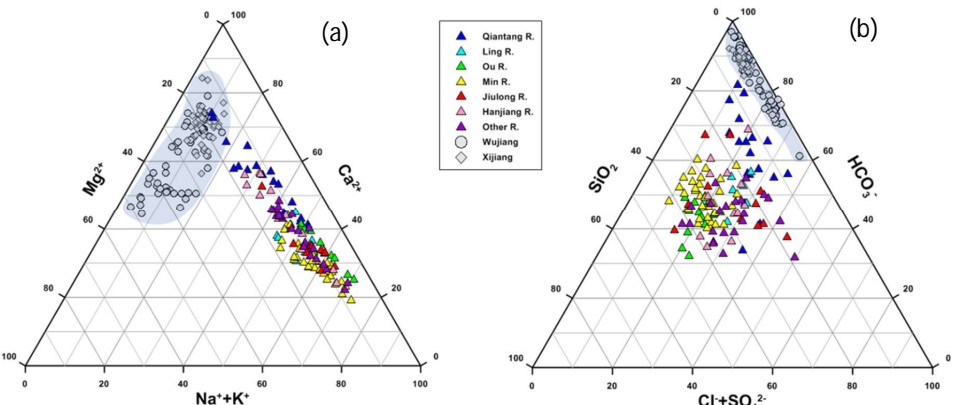

Fig. 2. Ternary diagrams showing cations (a), anions and dissolved $SiO_2$ (b)

compositions of river waters in the SECRB. Chemical compositions from case studies

of rivers draining carbonate rocks are also shown for comparison (data from Han and

Liu 2004; Xu and Liu 2007, 2010)

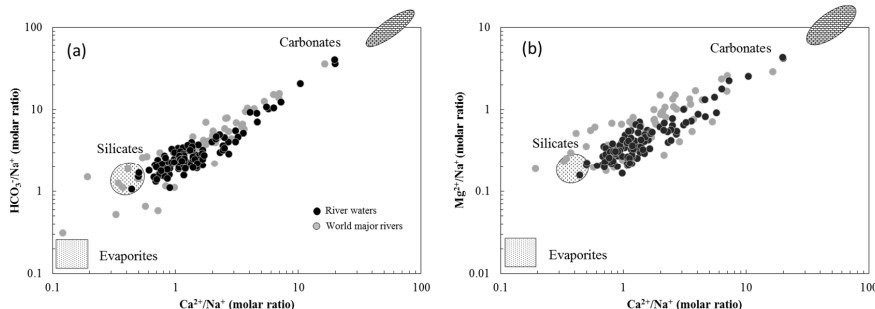

Fig. 3. Mixing diagrams using Na-normalized molar ratios: $HCO_3^-/Na^+$ *vs.* $Ca^{2+}/Na^+$

(a) and $Mg^{2+}/Na^+$ *vs.* $Ca^{2+}/Na^+$ (b) for the SECRB, showing a mixing line between

silicate and carbonate end-members. Data for world major rivers are also plotted for

comparison (data from Gaillardet et al. 1999).



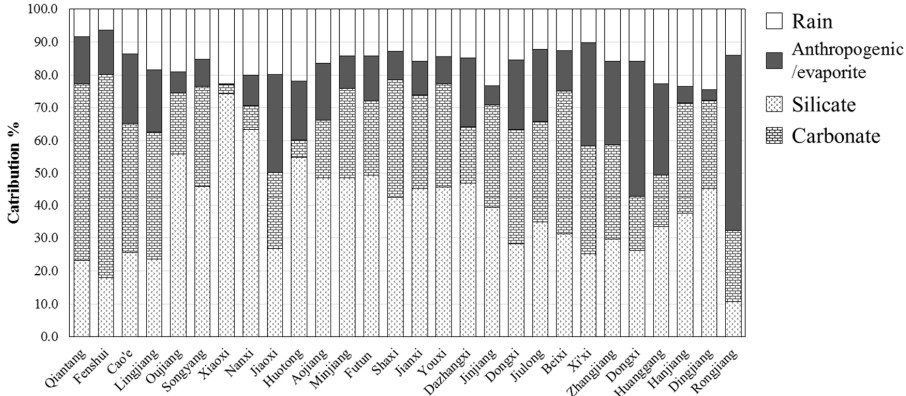

Fig. 4. Calculated contributions (in %) from the different reservoirs to the total

cationic load for major rivers and their main tributaries in the SECRB. The cationic

load is equal to the sum of $Na^+$, $K^+$, $Ca^{2+}$ and $Mg^{2+}$ from the different reservoirs.

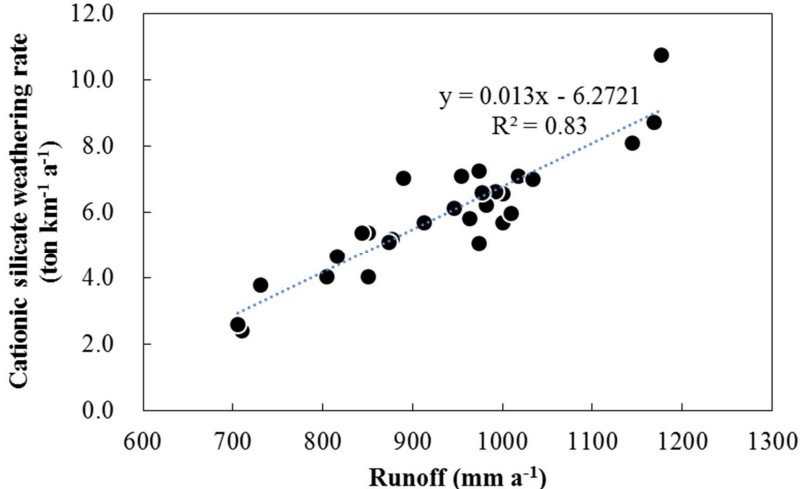

Fig. 5. Plots of the cationic-silicate weathering rate ($Cat_{sil}$) vs. runoff for the SECRB,

showing that runoff has a strong control on chemical weathering rates of silicates.





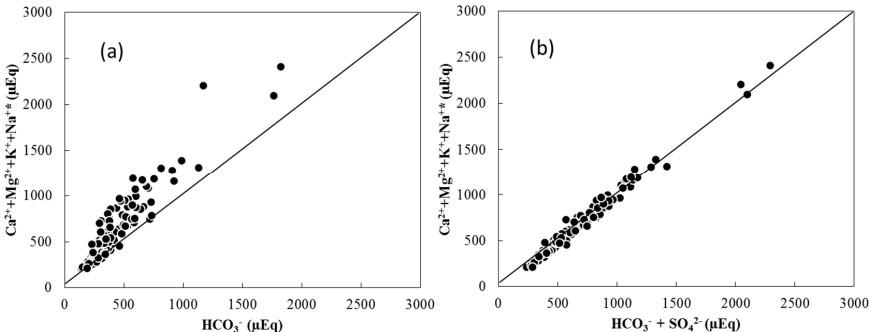

Fig. 6. Plots of total cations derived from carbonate and silicate weathering *vs.* $HCO_3^-$
(a) and $HCO_3^- + SO_4^{2-}$ (b) for river waters in the SECRB.

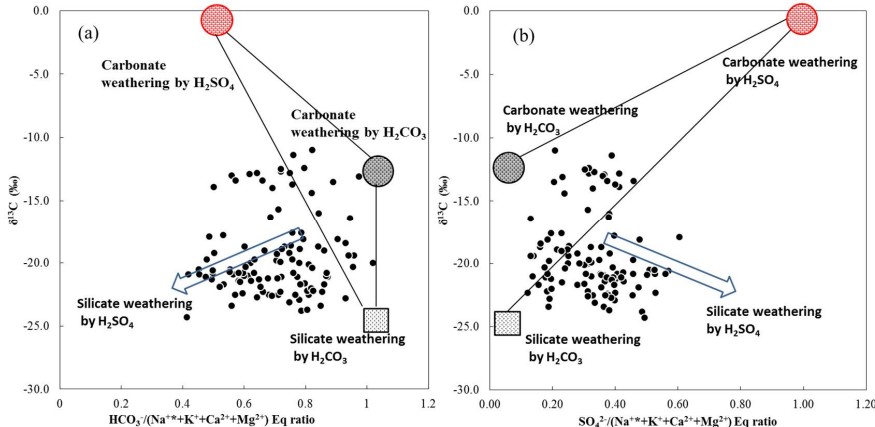

Fig. 7. $\delta^{13}C_{DIC}$ *vs.* $HCO_3/(Na*+K+Ca+Mg)$ (a) and $SO_4/(Na*+K+Ca+Mg)$ Eq ratio

(b) in river waters draining the SECRB. The plot show that most waters deviate from

the three endmember mixing area (carbonate weathering by carbonic acid and sulfuric

acid and silicate weathering by carboinic acid), illustrating the effect of sulfuric acid

on silicate weathering.