# Peer review of "Geochemistry of the dissolved loads of rivers in Southeast Coastal Region, China: Anthropogenic impact on chemical weathering and carbon sequestration"

_Biogeosciences, 2018_

## Referee Comment (RC1) · Anonymous Referee #1 · 10 Apr 2018

The Ms explored chemical weathering drawdown CO2 rates, major ion sources, and contribution of anthropogenic acids in the chemical weathering in a most severe acid rain impacted region, China. This is interesting, and the Ms is well structured and well written overall. The ms could be improved with consideration as follows

The field trip was conducted in the high-flow period. Whether is one hydrological sampling representative or can it represent a hydrological year, which must be explicated.

Alkalinity is titrated using HCl, while in the dataset of Table there is no alkalinity. I guess

that the HCO3 is from Alk, is it right? If yes, please demonstrate how to calculate the HCO3.

Authors referred many studies of rock chemical weathering, while several studies in Asia, such as Han River in the Yangtze and Mekong River in the Southeast Asian were ignored.

Authors should inform the extent of CO2 consumption rate in this study in contrast to the world rivers, particularly Asian rivers and highly-impacted rivers.

I have noted that the references is mostly old, some new citations should be included.

L 65 Change stronger to intense

L 138 How many samples?

L232-L233 Very high proportion of SO4 and NO3 is from atmosphere, if correct, does it mean the estimated CO2 consumption rate is still overestimated because of contribution of HNO3?

L393-394 Please could you supply the chemical equations for these weathering by HCO3, H2SO4 or both HCO3 and H2SO4. This will be helpful for readers to quantify the end-members.

L477 No year for this citation

Fig. 5. Please add p value

---

## Short Comment (SC1) · 14 Apr 2018

General comments: This study estimated the chemical weathering rates and atmospheric CO2 consumption rates in the coastal catchments of SE China, based on the chemistry and isotopes of dissolved inorganic carbon in the coastal rivers. The most important finding of this work is the sulfuric acid plays an unignored role in chemical weathering of carbonate and silicate rocks, which has to be more carefully considered in the calculation of weathering rates and carbon cycling in the catchments where strong human activities occur. Overall, the paper was well organized and structured,

and the major research conclusion will increase our better understanding of weathering process in river catchments. I basically agree with the major research findings of these study based on the high data quality and interpretation. My minor concern is about the influence of extreme climate events on weathering processes. As some studies suggest, the SE China is subject to strong typhoon impact every year, which could significantly alter the river water chemistry and probably weathering process in the catchments during typhoon season. This impact could not be ignored in the discussion part.

More specific comments and suggestions: 1) L97-100: How did you define the sizes (small, medium, large) of these different rivers in SE China? Based on their catchment areas, lengths or riverwater and sediment discharges? 2) On River settings: I suggest this part should include the mean water (and sediment) discharges of these rivers investigated. 3) L109: Data source? 4) L126: No influence of the Pacific Plate? 5) L141-142: To my knowledge, the estuaries and lower reaches of most of these river studied are subject to strong tidal influence. Based on the sampling locations on the map of Figure 1, it seems that some riverwater samples were taken much closer to the river mouths. Please make sure that all these samples were not influenced by tidal-pumped sea water, or you have some special method to correct this kind of influence. 6) L177: change to "Compared" 7) L181-182: Where are these rivers located? 8) L248-249: Considering the sizes of these rivers investigated, it may be more reasonable to compare them with those small- or medium-sized river systems. 9) L271: Do you mean the source rock types? To my knowledge, the tectonic settings of these rivers are much different. The climate regimes and anthropogenic activities as well are also much variable among these river catchments. 10) L322-324: What are the major reasons for the different silicate weathering rates observed in these river catchments? If the monsoon climate dominates the weathering process, the Xijiang in the southernmost should have the highest silicate weathering rates while the Huanghe in the northernmost has the lowest? 11) L401-402: How about the influence of seawater intrusion into the lower reaches of these rivers? 12) On the spelling of river names: It

always should keep in consistence in the text, figures and tables, e.g. Min, Jin, Han, Jiulong rivers, not "Minjiang, Jinjiang, Hanjiang. . ." 13) Table 1: The full names of TZ, EC, NICB and TDS should be given with the table. It's better to include the localities of these riverwater samples. 14) Table 3: Sources of riverwater discharges and runoff? 15) Figure 1: You'd better to mark the major names of rivers, and geographic localities, and tectonic units you mentioned in the text, e.g. Huanghe, Cathaysia and Yangtze blocks, Zhejiang and Fujian Provinces. 16) Figure 4: Wrong spelling of "Contribution" in Y axis. Add a name of "Rivers" to X axis. The spelling of river names should be keep in consistence. 17) On all figures: The fonts used in the diagrams should be consistent.

---

## Referee Comment (RC3) · Anonymous Referee #3 · 20 Apr 2018

Geochemistry of the dissolved loads of rivers in Southeast Coastal Region, China: Anthropogenic impact on chemical weathering and carbon sequestration, by Wenjing Liu et al.,

Many papers on dissolved loads in rivers have been published, but the papers about anthropogenic impacts on chemical weathering and carbon sequestration are rare. Thus this is an interesting paper. Used the water chemistry data measured in many rivers in the Southeast coastal region of China, Liu et al. presented their study on geochemistry of the dissolved loads in the region with severe acid rain impacts. They

sampled over 100 sites in the high-flow period in 2010, and employed the chemical compositions and carbon isotope ratio to quantify the associated atmospheric CO2 consumption rates and the contribution of anthropogenic acids. This study found that sulfuric acid played an important role in in chemical weathering, and acid deposition should be considered in studies of chemical weathering and associated CO2 consumption.

In addition, this paper provides a valuable dataset on the water chemistry which can be used for carbon fluxes study. Thus, this paper fits well into the theme of this special volume on carbon fluxes in Asian river. I recommend to accept this manuscript after some minor revision.

Line 215-217, when the authors discuss the source of Cl-, they say "In pristine areas, the concentration of Cl- in river water is assumed to be entirely derived from the atmosphere, provided that the contribution of evaporates is negligible". Please give a reference.

In fact it was found that ground water was an important source of Cl- for rivers in many regions of China such as the Yarlung Tsangpo basin on the Qinghai-Tibetan Plateau.

L232-L233 High proportion of SO4 and NO3 were found in the study area, but the discussion mainly focused on the SO4. What was the role of NO3 in the estimation of CO2 consumption rate?

Line 321-324 The authors made a comparison between the studied rivers in east coastal region and other major/large rivers in China such as Changjiang, Huanghe and Xijiang river. It will be good to have a forward discussion explaining the major reasons for the difference.

Line 386-387, "Carbonate rocks are generally derived from marine system and, typically, have $\delta$13C value close to zero", please add a reference

Table 1, how do you measure the HCO3? Are they calculated from the alkalinity?

please provide more info in the method section.

Fig. 5. Please provide the p value.

Other minor comments:

Line 72-74 the sentence is not well structured, please re-phrase.

Line 195 lack space between "%" and "of"
* * *

---

## Author Comment (AC1) · 29 Apr 2018

1. The field trip was conducted in the high-flow period. Whether is one hydrological sampling representative or can it represent a hydrological year, which must be explicated.

Reply: The river water of the southeast coastal rivers is mainly recharged by rain, and the amount of precipitation in high-flow season accounts for more than 70% of the annual precipitation in the area. During the high-flow season, the abundant water

recharging facilitates the weathering product entering river system. However, during the low-flow period, the ground water contribution to the surface water might be greater and overprint the real weathering information in river system, which would bring more inaccuracies to the weathering and CO2 consumption estimation. Therefore, it is more representative to investigate the rock weathering during the high-flow season in the subtropical monsoon climate watersheds.

2. Alkalinity is titrated using HCl, while in the dataset of Table there is no alkalinity. I guess that the HCO3 is from Alk, is it right? If yes, please demonstrate how to calculate the HCO3.

Reply: The content of HCO3- rather than alkalinity is titrated using HCl. We have made this point more clearly in the attached revision in the supplement.

3. Authors referred many studies of rock chemical weathering, while several studies in Asia, such as Han River in the Yangtze and Mekong River in the Southeast Asian were ignored.

Reply: According to the RC, we have cited these studies in the attached revision in the supplement.

4. Authors should inform the extent of CO2 consumption rate in this study in contrast to the world rivers, particularly Asian rivers and highly-impacted rivers.

Reply: According to the RC, we have compared the CO2 consumption rates of SECRB to the major rivers in the world and Asian. Please find it in Lines 349-360 in the attached revision in the supplement.

5. I have noted that the references is mostly old, some new citations should be included.

Reply: We have added recent studies in both the introduction and the discussion sections in the revised version attached in the supplement.

6. L 65 Change stronger to intense

Reply: It is revised in the attached revision in the supplement.

7. L 138 How many samples?

Reply: We have added the number of samples in the revision. Please find it in line 141 in the attached revision in the supplement.

8. L232-L233 Very high proportion of $SO_4$ and $NO_3$ is from atmosphere, if correct, does it mean the estimated $CO_2$ consumption rate is still overestimated because of contribution of $HNO_3$?

Reply: Yes, we do think the N deposition also plays a role in rock weathering and have impacts on $CO_2$ consumption. However, the $NO_3^-$ source in river is more complicated, e.g. atmospheric deposition, fertilizer, industry and urban waste water, as well as nitrification and denitrification. Without more information for the above source and more tools to distinguish the different $NO_3^-$ source, we could not address more conclusion on the effect of $HNO_3$ on $CO_2$ consumption rate. It would be an interesting theme for further study in this area.

9. L393-394 Please could you supply the chemical equations for these weathering by $HCO_3$, $H_2SO_4$ or both $HCO_3$ and $H_2SO_4$. This will be helpful for readers to quantify the end-members.

Reply: The chemical equations for carbonate and silicate weathering by $HCO_3$ and $H_2SO_4$ have been repetitively mentioned in many previous basin scale weathering studies (e.g. Li et al., 2008; Spence, and Telmer, 2005; Chetelat et al., 2008; Xu and Liu, 2010). In addition, we discussed the $\delta 13C$ isotopic composition of the end-members in lines 390-406. For the condense of the whole manuscript, we did not provide the chemical equations for carbonate and silicate weathering by $HCO_3$ and $H_2SO_4$.

10. L477 No year for this citation

11. Fig. 5. Please add p value

Reply: We have added p value in Fig. 5 in the attached revision in the supplement.

Please also note the supplement to this comment:
https://www.biogeosciences-discuss.net/bg-2018-109/bg-2018-109-AC1-supplement.pdf

---

## Author Comment (AC2) · 29 Apr 2018

General comments and replies:

This study estimated the chemical weathering rates and atmospheric CO2 consumption rates in the coastal catchments of SE China, based on the chemistry and isotopes of dissolved inorganic carbon in the coastal rivers. The most important finding of this work is the sulfuric acid plays an unignored role in chemical weathering of carbonate and silicate rocks, which has to be more carefully considered in the calculation of weathering rates and carbon cycling in the catchments where strong human activities occur. Overall, the paper was well organized and structured, and the major research conclusion will increase our better understanding of weathering process in river catchments. I basically agree with the major research findings of these study based on the high data quality and interpretation. My minor concern is about the influence of extreme climate events on weathering processes. As some studies suggest, the SE China is subject to strong typhoon impact every year, which could significantly alter the river water chemistry and probably weathering process in the catchments during typhoon season. This impact could not be ignored in the discussion part.

Reply: Yes, extreme climate events do have impacts on weathering processes, especially the geochemistry signals of river water. The impact could be generally temporal and regional. In the sampling period, typhoon "Chanthu" have landed on Guangdong province in July 22, 2010. However its major impaction area is Guangdong, Guangxi and Yunnan province, which are relatively far from our study basins. So, extreme climate events are not considered in this study. To be more cautious, in this revision, we applied for the open access to the Annual Hydrological Report P. R. China and have got more detailed data from different hydrology observation sites.

More specific comments and suggestions:

1) L97-100: How did you define the sizes (small, medium, large) of these different rivers in SE China? Based on their catchment areas, lengths or riverwater and sediment discharges?

Reply: The sizes of the rivers are based on the length and the drainage area. We have added some information in the revision. Pls find it in lines 101-102 in the attached revision in the supplement.

2) On River settings: I suggest this part should include the mean water (and sediment) discharges of these rivers investigated.

Reply: As there are many rivers, we did not provide the discharge data one by one in the main text. For the refinement of the article, the discharge and basin area information are provided in table 3 in the attached revision in the supplement..

3) L109: Data source?

Reply: It is calculated by the population and the administrative area of these three province.

4) L126: No influence of the Pacific Plate?

Reply: The formation of Yanshanian granitic rocks are closely related to multiple collision events between Cathaysia and Yangtze blocks and Pacific plate. We have added it in the attached revision in the supplement..

5) L141-142: To my knowledge, the estuaries and lower reaches of most of these river studied are subject to strong tidal influence. Based on the sampling locations on the map of Figure 1, it seems that some river water samples were taken much closer to the river mouths. Please make sure that all these samples were not influenced by tidal pumped sea water, or you have some special method to correct this kind of influence.

Reply: Yes, the estuary samples might be affected by seawater. We selected the sampling sites carefully to make it as far as possible from the tidal impacted area and also we avoid sampling during tidal period. In addition, we double checked the water chemistry data before drafting the manuscript and rule out the samples might be contaminated by seawater.

6) L177: change to "Compared"

Reply: It is modified in the revision. pls find it in the supplement.

7) L181-182: Where are these rivers located?

Reply: The location of the rivers are given in the in the attached revision in the supplement.

8) L248-249: Considering the sizes of these rivers investigated, it may be more reasonable to compare them with those small- or medium-sized river systems.

Reply: Data from Gaillardet et al. (1999) are cited here as global typical end-members and variation trend, to put the SECRB in a big picture instead of comparison. To avoid misunderstandings, "for comparison" was removed in the attached revision in the supplement.

9) L271: Do you mean the source rock types? To my knowledge, the tectonic settings of these rivers are much different. The climate regimes and anthropogenic activities as well are also much variable among these river catchments.

Reply: Yes, we have modified 'geological' to 'lithological' in the attached revision in the supplement.

10) L322-324: What are the major reasons for the different silicate weathering rates observed in these river catchments? If the monsoon climate dominates the weathering process, the Xijiang in the southernmost should have the highest silicate weathering rates while the Huanghe in the northernmost has the lowest?

Reply: Silicate weathering are complicated and affected by lithological setting, temperature and precipitation, etc. Silicate weathering rates in southeast coastal area is higher than the Xijiang and Huanghe but lower than Changjiang basin is the complicated results of silicate dominated bedrock (compared with Xijiang), high MAT and high runoff (compared with Huanghe and Changjiang basin). We added some discussion with rivers in China and the world in the following section in the attached revision in the supplement.

11) L401-402: How about the influence of seawater intrusion into the lower reaches of these rivers?

Reply: The sampling sites in the lower reaches are selected carefully: as far as possible to the estuaries to avoid the contamination of seawater. In addition, we carefully checked the water chemistry data before drafting the manuscript. The easily contaminated ions by seawater such as Cl-, Na+ and SO42- in the lower reach samples are in the normal range of fresh water.

12) On the spelling of river names: It always should keep in consistence in the text, figures and tables, e.g. Min, Jin, Han, Jiulong rivers, not "Minjiang, Jinjiang, Hanjiang"

Reply: We have improved this in the revision. pls find them in the supplement.

13) Table 1: The full names of TZ, EC, NICB and TDS should be given with the table. It's better to include the localities of these riverwater samples.

Reply: We added the full names of TZ, EC, NICB and TDS in table1. The localities of the samples are given in Fig. 1. Pls find all of these in the attached revision in the supplement. For the condense of the table, we did not give the longitude and latitude information in it.

14) Table 3: Sources of riverwater discharges and runoff?

Reply: The data of basin area and annual discharge are from Annual Hydrological Report P. R. China, 2010, vol (7). The runoff was calculated by data of annual discharge and basin area. We added the information in the revision. pls find it in line 303 and the references.

15) Figure 1: You'd better to mark the major names of rivers, and geographic localities, and tectonic units you mentioned in the text, e.g. Huanghe, Cathaysia and Yangtze blocks, Zhejiang and Fujian Provinces.

Reply: We have modified this in the Fig. 1 in the revision. pls find them in the supplement.

16) Figure 4: Wrong spelling of "Contribution" in Y axis. Add a name of "Rivers" to X axis. The spelling of river names should be keep in consistence.

17) On all figures: The fonts used in the diagrams should be consistent.

Reply for 16) and 17): We have improved this in the revision. pls find them in the supplement.

Please also note the supplement to this comment:
https://www.biogeosciences-discuss.net/bg-2018-109/bg-2018-109-AC2-supplement.pdf

─────────────────────────────

**Supplement:**

[revised manuscript text omitted]

---

## Author Comment (AC3) · 29 Apr 2018

1. Line 215-217, when the authors discuss the source of Cl-, they say "In pristine areas, the concentration of Cl- in river water is assumed to be entirely derived from the atmosphere, provided that the contribution of evaporates is negligible". Please give a reference. In fact it was found that ground water was an important source of Cl- for rivers in many regions of China such as the Yarlung Tsangpo basin on the Qinghai-Tibetan Plateau.

Reply: The reference for using the lowest Cl- in river water as the atmospheric input has been added in the attached revision in the supplement. As the reviewer suggested, the Qinghai-Tibetan Plateau and arid area, groundwater play as an important source for Cl-. However, in humid and hot area alike Southeast China, no salt-bearing rocks was found there. In addition, river water is mainly recharged by rain, but groundwater contribution is far more less than arid area. So, groundwater impact on river Cl- is not considered in this study.

2. L232-L233 High proportion of $SO_4$ and $NO_3$ were found in the study area, but the discussion mainly focused on the $SO_4$. What was the role of $NO_3$ in the estimation of $CO_2$ consumption rate?

Reply: Yes, we do think the existing of N deposition will make the $CO_2$ consumption estimation higher than ignore the role of $HNO_3$ during weathering. However, the source of riverine $NO_3^-$ is complicated, e.g. atmospheric deposition, fertilizer, industry and urban waste water, as well as the effect of nitrification and denitrification. Due to lack of information for the above source and more tools to distinguish the different $NO_3^-$ source, we could not address more conclusion on the effect of $HNO_3$ on $CO_2$ consumption rate within trustful errors. It would be an interesting theme for further study in this area though.

3. Line 321-324, The authors made a comparison between the studied rivers in east coastal region and other major/large rivers in China such as Changjiang, Huanghe and Xijiang river. It will be good to have a forward discussion explaining the major reasons for the difference.

Reply: Silicate weathering are complicated and affected by lithological setting, temperature and precipitation, etc. Silicate weathering rates in southeast coastal area is higher than the Xijiang and Huanghe but lower than Changjiang basin is the complicated results of silicate dominated bedrock (compared with Xijiang), high MAT and high runoff (compared with Huanghe and Changjiang basin). We added some discussion with rivers in Asia and the world in the following section in the attached revision in the supplement.

4. Line 386-387, "Carbonate rocks are generally derived from marine system and, typically, have 13C value close to zero", please add a reference

Reply: The reference has been added in the attached revision in the supplement.

5. Table 1, how do you measure the HCO3? Are they calculated from the alkalinity? Please provide more info in the method section.

Reply: The content of HCO3- rather than alkalinity is titrated using HCl. We have made this point more clearly in the attached revision in the supplement.

6. Fig. 5. Please provide the p value.

Reply: P value is provided in the attached revision in the supplement.

Other minor comments:

Line 72-74 the sentence is not well structured, please re-phrase.

Reply: We have re-phrase it in the revision. pls find it in the attached revision in the supplement.

Line 195 lack space between "%" and "of"

Reply: Modified in the attached revision in the supplement.

Please also note the supplement to this comment:
https://www.biogeosciences-discuss.net/bg-2018-109/bg-2018-109-AC3-supplement.pdf

[Figure]

**Supplement:**

[revised manuscript text omitted]

---

## Short Comment (SC2) · 30 Apr 2018

Comment on Geochemistry of the dissolved loads of rivers in Southeast Coastal Region, China: Anthropogenic impact on chemical weathering and carbon sequestration

Liu et al. report here some interesting data and interpretations on chemical weathering in southern china; using classical geochemical analysis, they were able to quantify the respective contributions of different weathering reactions in the watersheds, including those impacted by humans through acid rain deposition. The paper fits well within

the scope of the BG special issue and is based on appropriate methods. However, I found important anomalies concerning the way this paper refers to previous works by the same group on the same topics and in the same region, which makes its originality questionable. Indeed, the submitted paper presents many similarities with a paper published in 2016 (Liu et al. (2016) Water geochemistry of the Qiantangjiang River, East China: Chemical weathering and $CO_2$ consumption in a basin affected by severe acid deposition, Journal of Asian Earth Sciences Volume 127, 246-256), although the 2016 paper IS NOT CITED HERE. From my brief analysis of the two papers, I understood they were based on different datasets in different watersheds (although it is not clear if the dataset published in 2016 is included in the BGD paper or not, and why not). However the construction of the two papers is identical with many similarities in the text. In addition, figs. 2, 3, 4 and 6 in the submitted BGD paper (4 figures on 7 in total) are very similar from those in Liu et al. (2016) although with different data. In order to respect good practice in publishing scientific work, similarities in figures and text should be minimized, and the submitted paper should refer to previous similar works in the same region, citing and incorporating the information already available in the new paper, and extending its conclusion to a broader context. Since the approach and conclusions of the two papers are very similar, it is very odd that the Liu et al. (2016) is not cited and data from the Qiantangjiang River are not incorporated here or at least discussed in comparison with this new dataset.

Gwenaël Abril, BG associate editor, Editor of the special issue "Human impacts on carbon fluxes in Asian river systems"

---

## Author Comment (AC4) · 1 May 2018

Dear Dr. Gwenaël Abril,

Thanks for your interests in our work. Yes, this work is a further study after the previous work in the Qiantang River. However, we think some progresses were achieved in this work compared with the Qiantang river work.

The main goal of the manuscript is to evaluate acid deposition effect on silicate chemical weathering and CO2 sequestration in one of the three major acid rain areas in

China (even in east Asia) - Southeast Coastal Region. Dataset in this work incorporates all the major river basins there. This is not a simple attempt to increase sample numbers. As we know, the river basins in southeast coastal China have varied lithology, scale, runoff and anthropogenic background. Investigation in the river systems in the whole area would provide more accurate estimation of acid deposition impact on $CO_2$ consumption by chemical weathering in southeast China than just simply infer it with only one river basin like the Qiantang River there. In addition, the Qiantang river basin locates at the northwest corner of the target area, and is dominated by carbonate and detrital bedrock, while ingenious is the main rock type in the target area (Fig. 1 in the manuscript). It has been well documented in many weathering researches that lithology serves as an important (first order) control on weathering. So, it is necessary to carry out further investigation on the acid effect on silicate weathering and $CO_2$ consumption in the whole southeast coastal area after a case study of Qiantang river basin, and the conclusion would be more accurate and direct to meet with the topic of "Human impacts on carbon fluxes in Asian river systems" special issue.

Yes, the previous the Qiantang work should not be missed. We have 121 samples in this manuscript from the whole southeast coastal region, and 18 out of them is from the Qiantang river basin. We have noted them in table 1 and clarified the citation to avoid originality problem. Thanks a lot for your reviewing and kind reminding. Also, we have cited the paper and illustrate the motivation for this work base on our previous studies in the revised manuscript in the supplement (Line 86 to Line 104). About the data plots in the manuscript, figures of ternary diagrams, mixing diagrams with Na-normalized molar ratios, and contribution from the different reservoirs are commonly used procedures to identify the solute origin and their contribution.

Last but not least, after cautiously considering of the constructive comments, the role of another anthropogenic acid ($HNO_3$) played in the $CO_2$ consumption or at least the atmospheric $HNO_3$ input could be constrained in this study. So, we added the discussions of the atmospheric $HNO_3$ contribution during acid erosion of basin bedrock and revised the Fig. 6 in the attached revision in the supplement, to evaluate the anthropogenic impact (sulfuric and nitric acid) on the CO2 consumption of rock weathering. Hope these efforts will illustrate the relationship between our current and previous work, and make this work be considered as a possible contribution to the special issue.

Thanks for your time and work.

Sincerely yours,

Zhifang Xu on behalf of the Authors

Please also note the supplement to this comment:
https://www.biogeosciences-discuss.net/bg-2018-109/bg-2018-109-AC4-supplement.pdf

**Supplement:**

[revised manuscript text omitted]

---

## Referee Comment (RC4) · Anonymous Referee #2 · 2 May 2018

The revised paper made considerable improvement on the original version. I do not have further comments on this revised version.

---

## Author Comment (AC5) · 11 May 2018

Final Response to RC1:

Comments from Referees:

The Ms explored chemical weathering drawdown CO2 rates, major ion sources, and contribution of anthropogenic acids in the chemical weathering in a most severe acid rain impacted region, China. This is interesting, and the Ms is well structured and well written overall. The Ms could be improved with consideration as follows.

1. The field trip was conducted in the high-flow period. Whether is one hydrological sampling representative or can it represent a hydrological year, which must be explicated.

Author's response: The river water of the southeast coastal rivers is mainly recharged by rain, and the amount of precipitation in high-flow season accounts for more than 70% of the annual precipitation in the area. During the high-flow season, the abundant water recharging facilitates the weathering product entering river system. However, during the low-flow period, the ground water contribution to the surface water might be greater and overprint the weathering information in river system, which would bring more inaccuracies to the weathering and CO2 consumption estimation. Therefore, it could be more representative to investigate the rock weathering during the high-flow season in the subtropical monsoon climate watersheds in this study.

2. Alkalinity is titrated using HCl, while in the dataset of Table there is no alkalinity. I guess that the HCO3 is from Alk, is it right? If yes, please demonstrate how to calculate the HCO3.

Author's response: The content of HCO3- rather than alkalinity is titrated using HCl. We have made this point more clearly in the attached revision in the supplement.

3. Authors referred many studies of rock chemical weathering, while several studies in Asia, such as Han River in the Yangtze and Mekong River in the Southeast Asian were ignored.

Author's response: According to the RC, we have cited these studies in the attached revision in the supplement in both introduction and discussion sections.

4. Authors should inform the extent of CO2 consumption rate in this study in contrast to the world rivers, particularly Asian rivers and highly-impacted rivers.

Author's response: According to the RC, we have compared the CO2 consumption rates of SECRB to the major rivers in the world and Asian. Please find it in Lines

366-378 in the attached revision in the supplement.

5. I have noted that the references is mostly old, some new citations should be included.

Author's response: We have added recent studies in both the introduction and the discussion sections in the revised version attached in the supplement.

6. L 65 Change stronger to intense

Author's response: It is revised in the attached supplement.

7. L 138 How many samples?

Author's response: We have added the number of samples, please find it in line 155 in the attached revision in the supplement.

8. L232-L233 Very high proportion of SO4 and NO3 is from atmosphere, if correct, does it mean the estimated CO2 consumption rate is still overestimated because of contribution of HNO3?

Author's response: Yes, we do think the N deposition also plays a role in rock weathering and have impacts on CO2 consumption. However, the sources of NO3- in river waters are complicated, e.g. atmospheric deposition, fertilizer, industry and urban waste water, as well as nitrification and denitrification. Although it is difficult to determine the origin of nitrate in river waters, we can at least assume that nitrate from acid deposition is one of the providers of protons. We added the discussions about the effect of HNO3 in section 5.4, and recalculated the CO2 consumption in the SCERB. Please find them (from lines 381 to 450) in the attached revision.

9. L393-394 Please could you supply the chemical equations for these weathering by HCO3, H2SO4 or both HCO3 and H2SO4. This will be helpful for readers to quantify the end-members.

Author's response: The chemical equations for carbonate and silicate weathering by

HCO3 and H2SO4 have been repetitively mentioned in many previous basin scale weathering studies (e.g. Li et al., 2008; Spence, and Telmer, 2005; Chetelat et al., 2008; Xu and Liu, 2010). In addition, we discussed the $\delta$13C isotopic composition of the end-members in lines 414-430 in the attached revision. For the condensing of the whole manuscript, we did not provide the chemical equations for carbonate and silicate weathering by HCO3 and H2SO4.

10. L477 No year for this citation

Author's response: The year is at the end of the citation.

11. Fig. 5. Please add p value

Author's response: We have added p value (p<0.01) in Fig. 5 in the attached revision.

Please also note the supplement to this comment:
https://www.biogeosciences-discuss.net/bg-2018-109/bg-2018-109-AC5-supplement.pdf

**Supplement:**

[revised manuscript text omitted]
 carbonic acid), illustrating the effect of sulfuric and nitric acid on silicate weathering.

---

## Author Comment (AC7) · 11 May 2018

Final Response to RC3

Comments from Referees:

Geochemistry of the dissolved loads of rivers in Southeast Coastal Region, China: Anthropogenic impact on chemical weathering and carbon sequestration, by Wenjing Liu et al., Many papers on dissolved loads in rivers have been published, but the papers about anthropogenic impacts on chemical weathering and carbon sequestration are

rare. Thus this is an interesting paper. Used the water chemistry data measured in many rivers in the Southeast coastal region of China, Liu et al. presented their study on geochemistry of the dissolved loads in the region with severe acid rain impacts. They sampled over 100 sites in the high-flow period in 2010, and employed the chemical compositions and carbon isotope ratio to quantify the associated atmospheric CO2 consumption rates and the contribution of anthropogenic acids. This study found that sulfuric acid played an important role in in chemical weathering, and acid deposition should be considered in studies of chemical weathering and associated CO2 consumption. In addition, this paper provides a valuable dataset on the water chemistry which can be used for carbon fluxes study. Thus, this paper fits well into the theme of this special volume on carbon fluxes in Asian river. I recommend to accept this manuscript after some minor revision.

1. Line 215-217, when the authors discuss the source of Cl-, they say "In pristine areas, the concentration of Cl- in river water is assumed to be entirely derived from the atmosphere, provided that the contribution of evaporates is negligible". Please give a reference. In fact it was found that ground water was an important source of Cl- for rivers in many regions of China such as the Yarlung Tsangpo basin on the Qinghai-Tibetan Plateau.

Author's response: The reference has been added in the attached revision. As the reviewer suggested, the Qinghai-Tibetan Plateau and arid area, groundwater play as an important source for Cl-. However, in humid and hot area like Southeast China, no salt-bearing rocks was found there. In addition, river water is mainly recharged by rain, but groundwater contribution is far more less than arid area. So, groundwater impact on river Cl- is not considered in this study.

2. L232-L233 High proportion of SO4 and NO3 were found in the study area, but the discussion mainly focused on the SO4. What was the role of NO3 in the estimation of CO2 consumption rate?

Author's response: Yes, we do think the N deposition also plays a role in rock weathering and have impacts on CO2 consumption. However, the sources of NO3- in river waters are complicated, e.g. atmospheric deposition, fertilizer, industry and urban waste water, as well as nitrification and denitrification. Although it is difficult to determine the origin of nitrate in river waters, we can at least assume that nitrate from acid deposition is one of the providers of protons. We added the discussions about the effect of HNO3 in section 5.4, and recalculated the CO2 consumption in the SCERB. Please find them (from lines 381 to 450) in the attached revision.

3. Line 321-324, The authors made a comparison between the studied rivers in east coastal region and other major/large rivers in China such as Changjiang, Huanghe and Xijiang river. It will be good to have a forward discussion explaining the major reasons for the difference.

Author's response: Silicate weathering are complicated and affected by lithological setting, temperature and precipitation, etc. Silicate weathering rates in southeast coastal area is higher than the Xijiang and Huanghe but lower than Changjiang basin is the complicated results of silicate dominated bedrock (compared with Xijiang), high MAT and high runoff (compared with Huanghe and Changjiang basin). We added some discussion with rivers in Asia and the world in the following section (section 5.4) in the attached revision in the supplement.

4. Line 386-387, "Carbonate rocks are generally derived from marine system and, typically, have 13C value close to zero", please add a reference

Author's response: The reference has been added in the attached revision in the supplement.

5. Table 1, how do you measure the HCO3? Are they calculated from the alkalinity? Please provide more info in the method section.

Author's response: The content of HCO3- rather than alkalinity is titrated using HCl.

We have made this point more clearly in the attached revision in the supplement.

6. Fig. 5. Please provide the p value.

Author's response: P value is provided in the attached revision (p<0.01) in the supplement.

Other minor comments from referee 3:

Line 72-74 the sentence is not well structured, please re-phrase.

Author's response: We have re-phrase it in the revision. pls find it in the attached revision in the supplement.

Line 195 lack space between "%" and "of"

Author's response: Modified in the attached revision in the supplement.

Please also note the supplement to this comment:
https://www.biogeosciences-discuss.net/bg-2018-109/bg-2018-109-AC7-supplement.pdf

―――――――――――――――――――

---

## Author Comment (AC10) · 16 May 2018

SC2: Liu et al. report here some interesting data and interpretations on chemical weathering in southern china; using classical geochemical analysis, they were able to quantify the respective contributions of different weathering reactions in the watersheds, including those impacted by humans through acid rain deposition. The paper fits well with in the scope of the BG special issue and is based on appropriate methods. However, I found important anomalies concerning the way this paper refers to previous works by the same group on the same topics and in the same region, which makes its

originality questionable. Indeed, the submitted paper presents many similarities with a paper published in 2016 (Liu et al. (2016) Water geochemistry of the Qiantangjiang River, East China: Chemical weathering and CO2 consumption in a basin affected by severe acid deposition, Journal of Asian Earth Sciences Volume 127, 246-256), although the 2016 paper IS NOT CITED HERE. From my brief analysis of the two papers, I understood they were based on different datasets in different watersheds (although it is not clear if the dataset published in 2016 is included in the BGD paper or not, and why not). However the construction of the two papers is identical with many similarities in the text. In addition, figs. 2, 3, 4 and 6 in the submitted BGD paper (4 figures on 7 in total) are very similar from those in Liu et al. (2016) although with different data. In order to respect good practice in publishing scientific work, similarities in figures and text should be minimized, and the submitted paper should refer to previous similar works in the same region, citing and incorporating the information already available in the new paper, and extending its conclusion to a broader context. Since the approach and conclusions of the two papers are very similar, it is very odd that the Liu et al. (2016) is not cited and data from the Qiantangjiang River are not incorporated here or at least discussed in comparison with this new dataset.

Author's response:

Thanks for your interests in our work. Yes, this work is a further study after the previous work in the Qiantang River. However, we think some progresses were achieved in this work compared with the previous work instead of just enlarging sampling area. First, the goal of the manuscript is to evaluate acid deposition effect on silicate chemical weathering and CO2 sequestration in one of the three major acid rain areas, the Southeast Coastal Region, in China (even in the world). Dataset in this work incorporates all the major river basins with different geology, human disturbing extent and climate background in the area (16 basins and Qiantang River is one of them). It is not a simple attempt to increase sample numbers. As we know and also pointed out by reviewer 2 in the comment, the river basins in southeast coastal China vary significantly
in tectonic background, lithology, scale, runoff and anthropogenic background. Investigation in the river systems in the whole area would provide more accurate estimation of acid deposition impact on chemical weathering and CO2 consumption in southeast China than just simply infer it with only one river basin like the Qiantang River there. In addition, the Qiantang river basin locates at the northwest corner of the target area, and is dominated by carbonate and detrital bedrock, while igneous rock is the main rock type in the target area (Fig. 1 in the manuscript). It has been well documented in many weathering researches that lithology serves as an important control on weathering. So, it is necessary to carry out further investigation on the acid effect on silicate weathering and CO2 consumption in the whole southeast coastal area after a case study of Qiantang river basin, and the conclusion would be more accurate and direct to meet with the topic of "Human impacts on carbon fluxes in Asian river systems" special issue.

Yes, the previous Qiantang work should not be missed. We have 121 samples in this manuscript from the whole southeast coastal region, in which 18 out of them is from the Qiantang river basin. We have noted them in table 1 and clarified the citation to avoid originality problem. Thanks a lot for your reviewing and kind reminding. Also, we have cited the paper and illustrate the motivation for this work base on our previous studies in the revised manuscript in the supplement (Line 97 to Line 108). About the data plots in the manuscript, figures of water chemistry ternary diagrams, mixing diagrams with Na-normalized molar ratios, and contribution from the different reservoirs are commonly used procedures to identify the solute origin and their contribution.

Last but not least, after cautiously considering of the constructive comments, the role of another anthropogenic acid (HNO3) played in the CO2 consumption or at least the atmospheric HNO3 input has been constrained in this study. So, we added the discussions of the atmospheric HNO3 contribution during acid erosion of basin bedrock in the attached supplement, to evaluate the anthropogenic impact (sulfuric and nitric acid deposition) on the CO2 consumption of rock weathering.

Hope these efforts will illustrate the relationship between our current and previous work, and make this work be considered as a possible contribution to the special issue.

Please also note the supplement to this comment:
https://www.biogeosciences-discuss.net/bg-2018-109/bg-2018-109-AC10-supplement.pdf

———————————————————

---

## Editor Comment (EC1) · V. V. S. S Sarma (Editor) · 8 Jun 2018

The authors well responded to the reviewers and interactive comments and manuscript is in the good shape. However, I have few following queries:

1. How did you avoid sampling during tidal period - Please explain in detail about this. This question was asked by Referee however the response is not satisfactory.

2. The field trip was conducted during high-flow period hence the discussion represents to high-flow period only and this must be explicitly mentioned in the text as well in the

title. ∼70% of annual precipitation occurs during high-flow period, and high weathering expected, the processes during dry period (low-flow) is different. (This issue was also raised by RC 1 - please explicitly mention about this).

3. How you could titrate only HCO3? Is it not TA and then calculated HCO3? It is not clear.

Once you include above said issues in the revised manuscript, it may be accepted.

---

## Editor Comment (EC2) · V. V. S. S Sarma (Editor) · 11 Jun 2018

The authors have responded to the all the comments raised by reviewers, interactive discussion and self satisfactorily and revised the manuscript accordingly. I therefore recommend for acceptance of this manuscript.

---

## Author Response (AR1)

**Author's Response**

**Point-by-point response to the reviews:**

**RC1:**

Comments from Referees:

The Ms explored chemical weathering drawdown $CO_2$ rates, major ion sources, and contribution of anthropogenic acids in the chemical weathering in a most severe acid rain impacted region, China. This is interesting, and the Ms is well structured and well written overall. The Ms could be improved with consideration as follows.

1. The field trip was conducted in the high-flow period. Whether is one hydrological sampling representative or can it represent a hydrological year, which must be explicated.

Author's response:

The river water of the southeast coastal rivers is mainly recharged by rain, and the amount of precipitation in high-flow season accounts for more than 70% of the annual precipitation in the area. During the high-flow season, the abundant water recharging facilitates the weathering product entering river system. However, during the low-flow period, the ground water contribution to the surface water might be greater and overprint the weathering information in river system, which would bring more inaccuracies to the weathering and $CO_2$ consumption estimation. Therefore, it could be more representative to investigate the rock weathering during the high-flow season in the subtropical monsoon climate watersheds in this study.

2. Alkalinity is titrated using HCl, while in the dataset of Table there is no alkalinity. I guess that the $HCO_3$ is from Alk, is it right? If yes, please demonstrate how to calculate the $HCO_3$.

Author's response:

The content of $HCO_3^-$ was calculated. Alkalinity was determined by phenolphthalein and methyl orange end point titration with dilute HCl. The HCl consumption volumes for phenolphthalein and methyl orange end point titration were used to calculate the $CO_3^{2-}$ and $HCO_3^-$. Actually, there were little phenolphthalein alkalinity for all the samples (i.e. the HCl consumption volume for phenolphthalein end point titration were almost zero). The method was given in the Sampling and analytical method section in the revised version.

3. Authors referred many studies of rock chemical weathering, while several studies in Asia, such as Han River in the Yangtze and Mekong River in the Southeast Asian were ignored.

Author's response:

According to the RC, we have cited these studies in the attached revision in both introduction and discussion sections.

4. Authors should inform the extent of $CO_2$ consumption rate in this study in contrast to the world rivers, particularly Asian rivers and highly-impacted rivers.

Author's response:

According to the RC, we have compared the $CO_2$ consumption rates of SECRB to the major rivers in the world and Asian. Please find it in Lines 366-379 in the attached revision in the supplement.

5. I have noted that the references is mostly old, some new citations should be included.

Author's response:

We have added recent studies in both the introduction and the discussion sections in the revised version.

6. L 65 Change stronger to intense

Author's response:

It is revised in the revision.

7. L 138 How many samples?

Author's response:

We have added the number of samples in the revision.

8. L232-L233 Very high proportion of $SO_4$ and $NO_3$ is from atmosphere, if correct, does it mean the estimated $CO_2$ consumption rate is still overestimated because of contribution of $HNO_3$?

Author's response:

Yes, we do think the N deposition also plays a role in rock weathering and have impacts on $CO_2$ consumption. However, the sources of $NO_3^-$ in river waters are complicated, e.g. atmospheric deposition, fertilizer, industry and urban waste water, as well as nitrification and denitrification. Although it is difficult to determine the origin of nitrate in river waters, we can at least assume that nitrate from acid deposition is one of the providers of protons. We added the discussions about the effect of $HNO_3$ in section 5.4, and recalculated the $CO_2$ consumption in the SCERB in the revision.

9. L393-394 Please could you supply the chemical equations for these weathering by $HCO_3$, $H_2SO_4$ or both $HCO_3$ and $H_2SO_4$. This will be helpful for readers to quantify the end-members.

Author's response:

The chemical equations for carbonate and silicate weathering by $HCO_3$ and $H_2SO_4$ have been repetitively mentioned in many previous basin scale weathering studies (e.g. Li et al., 2008; Spence, and Telmer, 2005; Chetelat et al., 2008; Xu and Liu, 2010). In addition, we discussed the $\delta^{13}C$ isotopic composition of the end-members in lines 426-441 in the attached revision. For the condensing of the whole manuscript, we did not provide the chemical equations for carbonate and silicate weathering by $HCO_3$ and $H_2SO_4$.

10. L477 No year for this citation

Author's response:

The year is at the end of the citation.

11. Fig. 5. Please add p value

Author's response:

We have added p value ($p<0.01$) in Fig. 5 in the attached revision.

**RC2:**

Comments from Referees:

This study estimated the chemical weathering rates and atmospheric CO2 consumption rates in the coastal catchments of SE China, based on the chemistry and isotopes of dissolved inorganic carbon in the coastal rivers. The most important finding of this work is the sulfuric acid plays an unignored role in chemical weathering of carbonate and silicate rocks, which has to be more carefully considered in the calculation of weathering rates and carbon cycling in the catchments where strong human activities occur. Overall, the paper was well organized and structured, and the major research conclusion will increase our better understanding of weathering process in river catchments. I basically agree with the major research findings of these study based on the high data quality and interpretation. My minor concern is about the influence of extreme climate events on weathering processes. As some studies suggest, the SE China is subject to strong typhoon impact every year, which could significantly alter the river water chemistry and probably weathering process in the catchments during typhoon season. This impact could not be ignored in the discussion part.

Author's response:

Yes, extreme climate events do have impacts on weathering processes, especially the geochemistry signals of river water. The impact could be generally temporal and regional. In the sampling period, typhoon "Chanthu" have landed on Guangdong province in July 22, 2010. However its major impaction area is Guangdong, Guangxi and Yunnan province, which are relatively far away from our study basins. So, extreme climate events are not considered in this study. To be more cautious, during the period of public discussion, we successfully applied for the open access to the Annual Hydrological Report P. R. China and have got more detailed data from different hydrology observation sites to get a more accurate estimation of weathering and $CO_2$ consumption fluxes.

More specific comments and suggestions:

1) L97-100: How did you define the sizes (small, medium, large) of these different rivers in SE China? Based on their catchment areas, lengths or riverwater and sediment discharges?

Author's response:

The sizes of the rivers are based on the length and the drainage area. We have added this information in the Natural setting of study area section in the revision.

2) On River settings: I suggest this part should include the mean water (and sediment) discharges of these rivers investigated.

Author's response:

As there are many rivers, we did not provide the discharge data one by one in the main text. For the condensing of the MS, the discharge and basin area information are provided in table 3 in the revision.

3) L109: Data source?

Author's response:

It is calculated by the population and the administrative area of these three provinces.

4) L126: No influence of the Pacific Plate?

Author's response:

After more investigation of the regional tectonic background, we have added Pacific plate in the introduction of Yanshanian granitic rocks formation collision events. It is the result of multiple collision events between Cathaysia, Yangtze blocks and Pacific plate. Pls find them in lines 140-144 in the attached supplement.

5) L141-142: To my knowledge, the estuaries and lower reaches of most of these river studied are subject to strong tidal influence. Based on the sampling locations on the map of Figure 1, it seems that some river water samples were taken much closer to the river mouths. Please make sure that all these samples were not influenced by tidal pumped sea water, or you have some special method to correct this kind of influence.

Author's response:

Yes, the estuary samples might be affected by seawater. To avoid this, first, we selected the sampling sites carefully to make it as far as possible from the tidal impacted area and also we avoid sampling during tidal period. In addition, we double checked the salinity and water chemistry data to rule out the samples might be contaminated by seawater.

6) L177: change to "Compared"

Author's response:

It is modified in the revision.

7) L181-182: Where are these rivers located?

Author's response:

The locations of the rivers are given in the attached revision..

8) L248-249: Considering the sizes of these rivers investigated, it may be more reasonable to compare them with those small- or medium-sized river systems.

Author's response:

Data from Gaillardet et al. (1999) are cited here as global typical end-members and variation trend, to put the SECRB in a big picture instead of comparison. To avoid misunderstandings, "for comparison" was removed in the attached revision in the revision.

9) L271: Do you mean the source rock types? To my knowledge, the tectonic settings of these rivers are much different. The climate regimes and anthropogenic activities as well are also much variable among these river catchments.

Author's response:

Yes, we have modified 'geological' to 'lithological' to avoid misunderstanding in the attached revision.

10) L322-324: What are the major reasons for the different silicate weathering rates observed in these river catchments? If the monsoon climate dominates the weathering process, the Xijiang in the southernmost should have the highest silicate weathering rates while the Huanghe in the northernmost has the lowest?

Author's response:

Silicate weathering are complicated and affected by lithological setting, temperature and precipitation, etc. Silicate weathering rates in southeast coastal area is higher than the Xijiang and Huanghe but lower than Changjiang basin is the complicated results of silicate dominated bedrock (compared with Xijiang), high MAT and high runoff (compared with Huanghe and Changjiang basin). We added some discussion with rivers in China and around the world in the following section (5.4 $CO_2$ consumption and the role of sulfuric acid) in the attached revision.

11) L401-402: How about the influence of seawater intrusion into the lower reaches of these rivers?

Author's response:

The sampling sites in the lower reaches are selected carefully: as far as possible to the estuaries to avoid the contamination of seawater. In addition, we carefully checked the salinity and water chemistry data before drafting the manuscript. The easily contaminated ions by seawater such as $Cl^-$, $Na^+$ and $SO_4^{2-}$ in the lower reach samples are in the normal range of fresh water.

12) On the spelling of river names: It always should keep in consistence in the text, figures and tables, e.g. Min, Jin, Han, Jiulong rivers, not "Minjiang, Jinjiang, Hanjiang"

Author's response:

We have improved this through the MS in the revision. pls find them in the supplement.

13) Table 1: The full names of TZ, EC, NICB and TDS should be given with the table. It's better to include the localities of these riverwater samples.

Author's response:

We added the full names of TZ, EC, NICB and TDS in table1. The localities of the samples are given in Fig. 1. Pls find them in the attached revision. For the condensing of the table, we did not include the longitude and latitude information in it.

14) Table 3: Sources of riverwater discharges and runoff?

Author's response:

The data of basin area and annual discharge are from Annual Hydrological Report P. R. China, 2010, vol (7). The runoff was calculated by annual discharge and basin area. We added the information in the revision. pls find it in line 329-331 and the references.

15) Figure 1: You'd better to mark the major names of rivers, and geographic localities, and tectonic units you mentioned in the text, e.g. Huanghe, Cathaysia and Yangtze blocks, Zhejiang and Fujian Provinces.

Author's response:

We have modified this in the Fig. 1 in the revision.

16) Figure 4: Wrong spelling of "Contribution" in Y axis. Add a name of "Rivers" to X axis. The spelling of river names should be keep in consistence.
17) On all figures: The fonts used in the diagrams should be consistent.

Author's response to comment 16) and 17):

We have improved this in the revision.

**RC3:**

Comments from Referees:

Geochemistry of the dissolved loads of rivers in Southeast Coastal Region, China: Anthropogenic impact on chemical weathering and carbon sequestration, by Wenjing

Liu et al., Many papers on dissolved loads in rivers have been published, but the papers about anthropogenic impacts on chemical weathering and carbon sequestration are rare. Thus this is an interesting paper. Used the water chemistry data measured in many rivers in the Southeast coastal region of China, Liu et al. presented their study on geochemistry of the dissolved loads in the region with severe acid rain impacts. They sampled over 100 sites in the high-flow period in 2010, and employed the chemical compositions and carbon isotope ratio to quantify the associated atmospheric $CO_2$ consumption rates and the contribution of anthropogenic acids. This study found that sulfuric acid played an important role in in chemical weathering, and acid deposition should be considered in studies of chemical weathering and associated $CO_2$ consumption. In addition, this paper provides a valuable dataset on the water chemistry which can be used for carbon fluxes study. Thus, this paper fits well into the theme of this special volume on carbon fluxes in Asian river. I recommend to accept this manuscript after some minor revision.

1. Line 215-217, when the authors discuss the source of $Cl^-$, they say "In pristine areas, the concentration of $Cl^-$ in river water is assumed to be entirely derived from the atmosphere, provided that the contribution of evaporates is negligible". Please give a reference. In fact it was found that ground water was an important source of Cl- for rivers in many regions of China such as the Yarlung Tsangpo basin on the Qinghai-Tibetan Plateau.

Author's response:

The reference has been added in the attached revision. As the reviewer suggested, the Qinghai-Tibetan Plateau and arid area, groundwater play as an important source for $Cl^-$. However, in humid and hot area like Southeast China, no salt-bearing rocks was found there. In addition, river water is mainly recharged by rain, but groundwater contribution is far more less than arid area. So, groundwater impact on river $Cl^-$ is not considered in this study.

2. L232-L233 High proportion of $SO_4$ and $NO_3$ were found in the study area, but the discussion mainly focused on the $SO_4$. What was the role of $NO_3$ in the estimation of

$CO_2$ consumption rate?

Author's response:

Yes, we do think the N deposition also plays a role in rock weathering and have impacts on $CO_2$ consumption. However, the sources of $NO_3^-$ in river waters are complicated, e.g. atmospheric deposition, fertilizer, industry and urban waste water, as well as nitrification and denitrification. Although it is difficult to determine the origin of nitrate in river waters, we can at least assume that nitrate from acid deposition is one of the providers of protons. We added the discussions about the effect of $HNO_3$ in section 5.4, and recalculated the $CO_2$ consumption in the SCERB.

3. Line 321-324, The authors made a comparison between the studied rivers in east coastal region and other major/large rivers in China such as Changjiang, Huanghe and Xijiang river. It will be good to have a forward discussion explaining the major reasons for the difference.

Author's response:

Silicate weathering are complicated and affected by lithological setting, temperature and precipitation, etc. Silicate weathering rates in southeast coastal area is higher than the Xijiang and Huanghe but lower than Changjiang basin is the complicated results of silicate dominated bedrock (compared with Xijiang), high MAT and high runoff (compared with Huanghe and Changjiang basin). We added some discussion with rivers in Asia and the world in the following section (section 5.4) in the attached revision.

4. Line 386-387, "Carbonate rocks are generally derived from marine system and, typically, have 13C value close to zero", please add a reference

Author's response:

The reference has been added in the attached revision.

5. Table 1, how do you measure the HCO3? Are they calculated from the alkalinity? Please provide more info in the method section.

Author's response:

The content of $HCO_3^-$ was calculated. Alkalinity was determined by phenolphthalein and methyl orange end point titration with dilute HCl. The HCl consumption volumes for phenolphthalein and methyl orange end point titration were used to calculate the $CO_3^{2-}$ and $HCO_3^-$. Actually, there were little phenolphthalein alkalinity for all the samples (i.e. the HCl consumption volume for phenolphthalein end point titration were almost zero). The method was given in the Sampling and analytical method section in the revised version (Line 175-177).

6. Fig. 5. Please provide the p value.

Author's response:

P value is provided in the attached revision ($p<0.01$) in the supplement.

Other minor comments from referee 3:

Line 72-74 the sentence is not well structured, please re-phrase.

Author's response:

We have re-phrase it in the revision. pls find it in the attached revision in the supplement.

Line 195 lack space between "%" and "of"

Author's response:

Modified in the attached revision in the supplement.

**EC1:**

The authors well responded to the reviewers and interactive comments and manuscript is in the good shape. However, I have few following queries:

1. How did you avoid sampling during tidal period - Please explain in detail about this.

This question was asked by Referee however the response is not satisfactory.

Author's response:

To avoid tidal effect on the river estuary samples, the sampling sites were selected carefully based on the following consideration. First, the sampling locations for the river low reach samples were chosen as far as possible from the tidal impacted area, normally further than 30 km. Second, we checked the local daily tidal time and conducted the sampling of river low reach during low tide period in the sampling day. Third, we also checked the salinity of the water by using salinometer (WS202, China) before sampling in the field. In addition, we double checked data before drafting the manuscript to make sure the river sample are not contaminated by seawater (e.g. all the water chemistry features of the samples were within the normal range of fresh water). These were noted in the Sampling and analytical method section in the revised manuscript (Line 158-164 in the revision)

2. The field trip was conducted during high-flow period hence the discussion represents to high-flow period only and this must be explicitly mentioned in the text as well in the title. ~70% of annual precipitation occurs during high-flow period, and high weathering expected, the processes during dry period (low-flow) is different. (This issue was also raised by RC1 - please explicitly mention about this).
Author's response:

Yes, the processes in low-flow season might be different in some extent due to the hydrologic and temperature effect. Thanks for your kind reminder. We have explicit this point in the title and in the text in the revised version (Line 468-476 in the supplement of the response to EC1).

3. How you could titrate only HCO3? Is it not TA and then calculated HCO3? It is not clear.

Author's response:

Yes, the content of $HCO_3^-$ was calculated. Alkalinity was determined by phenolphthalein and methyl orange end point titration with dilute HCl. The HCl consumption volumes for phenolphthalein and methyl orange end point titration were used to calculate the $CO_3^{2-}$ and $HCO_3^-$. Actually, there were little phenolphthalein alkalinity for all the samples (i.e. the HCl consumption volume for phenolphthalein end point titration were almost zero). The method was given in the Sampling and analytical method section in the revised version (Line 175-177).

**A list of all relevant changes made in the manuscript:**

1. More river size information was provided in the revision in lines 111-115;

2. Total sampling numbers were given in the revised manuscript in line 152;

3. Procedures to avoid seawater contamination were detailed in lines 154-160 in the revision;

4. $HCO_3^-$ determination were more clearly clarified in the Sampling and analytical method section in the revised manuscript in lines 171-173;

5. More comparison between $CO_2$ consumption rates of SECRB and the major rivers in the world and Asian were conducted in the revision. Please find them in Lines 366-375 in the revision;

6. Both $H_2SO_4$ and $HNO_3$ were reconsidered in the revision (lines 387-395, 407-412, 441-449);

7. p value (p<0.01) was added in Fig. 5 in the revision;

8. The discharge data of the rivers were provided one by one in table 3 in the revision;

9. We added some discussion with rivers in Asia and the world in section (section 5.4) in the attached revision (Lines 375-377);

10. The reference for carbonate rock $\delta^{13}C$ has been added in the attached revision in Line 425;

11. We have discussed the river water weathering information in different hydrology season in in the revised version (Lines 456-463);

12. The hydrology season of the samples were added in the title;

**A marked-up manuscript version:**

[revised manuscript text omitted]
 carbonic acid), illustrating the effects of sulfuric and nitric acid on silicate weathering.